# ExpertLens: Activation steering features are highly interpretable

**Masha Fedzechkina**[*]                                          *mfedzechkina@apple.com*
*Apple*

**Eleonora Gualdoni**[*]                                          *e_gualdoni@apple.com*
*Apple*

**Sinead Williamson**[*]                                          *sa_williamson@apple.com*
*Apple*

**Katherine Metcalf**                                          *kmetcalf@apple.com*
*Apple*

**Skyler Seto**                                          *sseto@apple.com*
*Apple*

**Barry-John Theobald**                                          *bjtheobald@apple.com*
*Apple*

**Reviewed on OpenReview:** https://openreview.net/forum?id=FBIsN6RdYO

## Abstract

Activation steering methods in large language models (LLMs) have emerged as an effective way to perform targeted updates to enhance generated language without requiring large amounts of adaptation data. We ask whether the features discovered by activation steering methods are interpretable. We identify neurons responsible for specific concepts (e.g., "cat") using the "finding experts" method from research on activation steering and show that the ExpertLens, i.e., inspection of these neurons, provides insights about model representation. We find that ExpertLens representations are stable across models and datasets and closely align with human representations inferred from behavioral data, matching inter-human alignment levels. ExpertLens significantly outperforms the alignment captured by word/sentence embeddings and sparse autoencoder (SAE) features. By reconstructing human concept organization through ExpertLens, we show that it enables a granular view of LLM concept representation. Our findings suggest that ExpertLens is a flexible and lightweight approach for capturing and analyzing model representations.

## 1    Introduction

Recently, large language models (LLMs) have moved from being a scientific tool used in machine learning to being used by millions of users in everyday life in areas as different as coding (Barke et al., 2023; Jiang et al., 2024), tutoring (Yang et al., 2024; Scarlatos et al., 2025), and answering medical questions (Singhal et al.,

---

This work is a team effort. * indicates a core author.

2023). At the same time, there is a growing body of evidence suggesting that LLMs provide responses that are misaligned with expected societal norms and behaviors such as hallucinating information (Bubeck et al., 2023; Lin et al., 2022), generating toxic content (Gehman et al., 2020) or sensitivity to minor variations of a prompt (Errica et al., 2025). Such misaligned behaviors pose obstacles to safe and trustworthy deployment of LLMs in real-life scenarios and therefore developing methods to understand the inner workings of these models that give rise to such behaviors is becoming more pressing.

Several pre-existing interpretability methods have been adapted to work with LLMs. These primarily involve analyzing input-output relationships such as by prompting the the model in various ways to produce a particular behavior (Shaki et al., 2023), or using attributional methods such as Shapley values (Lundberg & Lee, 2017; Horovicz & Goldshmidt, 2024) that trace the model predictions. A natural extension of this is to also study intermediate representations, looking for interpretable patterns in the models' embeddings (Ettinger & Linzen, 2016; Sajjad et al., 2022). In recent years, the field of mechanistic interpretability (MI) has gained momentum. MI studies the fundamentals of model computation by identifying model components (such as features, neurons, layers, circuits) that are causally connected to the model's output (Geiger et al., 2021; Feng & Steinhardt, 2024; Vasileiou & Eberle, 2024; Bereska & Gavves, 2024). The focus on causal relationships and precise computation that transform the inputs into the outputs is the key differentiating factor between MI and other approaches.

A somewhat different family of approaches – activation steering – has also sought to find a causal role between the features they discover and model output (Rodriguez et al., 2025; Li et al., 2024; Rimsky et al., 2024). Unlike MI, activation steering is not focused on understanding the inner workings of the model but rather aims to discover approaches to control model behavior. These approaches typically involve two stages: first discovering the features that are *correlated* with the desired model behavior, and then manipulating these features to steer a model's generations towards that behavior (confirming a *causal* relationship). Activation steering methods typically require little data (a few hundred sentences usually suffice) and are relatively light-weight compared with many MI approaches, which makes them an attractive option for interpretability research at scale. However, we do not know if the features discovered by these methods are interpretable.

In this work, we provide an in-depth investigation of the features found by the "finding experts" activation steering method (Suau et al., 2023; 2024). This method identifies so-called *expert neurons*, i.e., the neurons most strongly associated with processing and understanding of a particular concept. We show that these neurons are stable across datasets and models (Sec. 4.2) and are causally connected to model generations (Sec. 4.1). More importantly, the dimensions they capture are meaningful to humans, providing an ExpertLens — a reliable method to test hypotheses about model representation (Sec. 4.3). We assess the use of ExpertLens as an interpretability tool in two tasks. First, we look at whether the similarity between ExpertLens representations for a pair of concepts is predictive of human-perceived concept similarity. Second, we use ExpertLens to reconstruct human conceptual structure (e.g., we ask if "dog", "cat", "cheetah", and "animal" share a consistent set of neurons) (Rosch, 1978). Finally, we study how ExpertLens representations develop through training (Sec. 4.4).

Our contributions are:

1. We show that ExpertLens reliably captures concept representations in LLMs and is stable across models and datasets.

2. We show that ExpertLens representations align closely with human representations matching alignment between humans, both at the level of concept similarity and in terms of concept organization, surpassing the levels of alignment detectable with approaches relying on embedding or sparse autoencoder (SAE) feature similarity.

3. We provide an analysis of how ExpertLens representations evolve with model training and model capacity.

Based on these contributions we conclude that our ExpertLens framework offers a lightweight but powerful option for interpretability of LLMs.

## 2 Related work

**Mechanistic interpretabily** MI is a the fast-growing field that seeks to reverse-engineer LLMs into human-interpretable components, revealing the neural and architectural pathways by which models process information (Geiger et al., 2021; Feng & Steinhardt, 2024; Vasileiou & Eberle, 2024). MI has provided a toolkit for model interpretability ranging from observational approaches that allow us to introspect model behavior such as probes (Belinkov, 2021), logit lens and its variants (nostalgebraist, 2020; Belrose et al., 2025), sparse autoencoders (Cunningham et al., 2023b) to interventional approaches that adopt a causal perspective on interpretability by intervening on model components such as activation, path or attribution patching (Meng et al., 2022; Goldowsky-Dill et al., 2023) or causal mediation analysis (Stolfo et al., 2023; Vig et al., 2020; Meng et al., 2022). Sparse autoencoders (Cunningham et al., 2023a), trained to discover salient and disentangled features, are widely used for steering and for interpretability where decomposing model activations into monosemantic features is advantageous.

**Activation steering** Activation steering is a class of methods that intervene on a generative model's activations to perform targeted updates for controllable generation (Rodriguez et al., 2025; Li et al., 2024; Rimsky et al., 2024; Wu et al., 2025). These methods have been successfully applied to a variety of problems from inducing a particular concept like "cat" (Wu et al., 2025; Suau et al., 2023) to reducing toxicity (Li et al., 2024; Suau et al., 2024) or sycophantic behavior (Rimsky et al., 2024) to understanding multilingual model capabilities (Riemenschneider & Frank, 2025; Sundar et al., 2025). Prior work has documented a causal connection between the features discovered by these methods and their role in model generations (Rodriguez et al., 2025; Li et al., 2024; Rimsky et al., 2024; Wu et al., 2025).

**Finding experts** We focus on one activation steering method — *finding experts* —introduced by (Suau et al., 2023) for several reasons. In terms of concept discovery, prior work (Suau et al., 2023) has shown that this approach can capture the neurons responsible for everyday concepts like "dog", which is the focus of this work and is able to distinguish the different senses of a homophone (e.g., "apple" as a fruit or company), suggesting that this method is able to pick up fine-grained semantic distinctions. Prior work has also established that expert neurons play a causal role in the generation of outputs semantically related to the concept the neurons encode. Specifically, Suau et al. (2023) and Faisal & Anastasopoulos (2023) show that activating the experts for concepts similar to the ones we are investigating (e.g., "dog" or "apple" or country names respectively) steers the model to generate text consistent with this concept. Suau et al. (2024) further show that suppressing the experts for toxicity generates less toxic text. Kojima et al. (2024) and Sundar et al. (2025) show that activating experts for a specific language (e.g., Spanish) leads multilingual models to produce text in that language in response to a neutral prompt.

Overall, work on activation steering demonstrates that it is possible to find expert neurons and use them to steer model generations into a desired direction. What we do not know is whether the set of identified expert neurons is stable across inputs (Sec.4.2) and, if so, whether these representations are interpretable, which is the focus of the current work.

## 3 Methods

### 3.1 Finding expert neurons

We follow the implementation of finding experts method in (Suau et al., 2023). We define a concept $c$ as a set of example sentences $N = N_c^+ + N_c^-$, where $N_c^+$ is a set of sentences that contain $c$ (henceforth *positive set*) and $N_c^-$ is a set of sentences that do not contain $c$ (henceforth *negative set*). Next, we obtain the activations $z_m^c = \{z_{m,i}^c\}_{i=1}^N$ for every neuron $m$ in the model in response to the inputs from both sets of sentences. $z_m^c$ is then treated as a prediction score for the presence of $c$, since we know the ground truth label. The performance of each neuron as a classifier for the concept (i.e., its expertise) is measured as the area under the precision-recall curve (AP) on this task. We calculate the AP score for all units in the MLP and attention layers. Formulated this way, the experts approach has several advantages: as discussed above, it is sensitive

to context and can distinguish different senses of a homophone; and it can be trivially extended to more abstract concepts like safety, toxicity, document style or other multi-word concepts.

We consider neurons with an AP score above a given threshold, $\tau$, for a concept to be expert neurons for that concept. $\tau$ can be thought of as quality of an expert neuron — the larger the value of $\tau$, the more expert a neuron is for a given concept. In our experiments, we consider a range of values for $\tau \in [0.5, 0.9]$ from a low to a high level of expertise. Additionally, we establish the causal connection between the expert neurons and the expression of the concept in the generations by activating the top-500 experts (i.e., by setting their activations to the mean value over the positive set).

## 3.2 Data

We assess the interpetabilty of ExpertLens representations by examining how patterns in these representations relate to perceived concept similarity in humans. We obtain human similarity judgments from two datasets: the MEN dataset (Bruni et al., 2014), which contains $3,000$ word pairs annotated with human-assigned similarity judgments crowd-sourced from Amazon Mechanical Turk, and the Semantic Priming Project (hereafter, SPP), a database of behavioral measures for related and unrelated word pairs (Hutchison et al., 2013). In this work, we focus on single-word concepts because we have the most reliable measures of human-perceived similarity for this type of concept. Our approach, however, can be trivially extended to multi-word concepts.

For each concept under consideration, we generate a set of sentences containing that concept. To ensure dataset diversity, half of each positive dataset is generated with a prompt eliciting story descriptions and half of the dataset is generated with a prompt eliciting factual descriptions of the target concept (the prompts, along with sample generations, are provided in App.A. The negative sets are sampled from the datasets for the remaining non-target concepts (e.g., if we are considering 1000 concepts, one of which is "cat", the negative set is sampled from 999 concepts excluding "cat"). As part of our initial exploration (Sec. 4.2), we experiment with three models of different performance levels: GPT-4 (OpenAI et al., 2024), Mistral-7b-Instruct-v0.2 (Jiang et al., 2023), and an internal 80b-chat model.

For the case study in ExpertLens concept organization and the exploration of model generations (Sec. 4.3 and Sec. 4.1), we manually generate lists of ten domains with four concepts per domain (e.g., the domain "animal" containing concepts "cat", "dog", "cheetah"', and "horse"; the full set of domains and concepts is provided in App. F.1). We choose not to use WordNet (Miller, 1994) — a lexical database of English — because of drawbacks identified in its hierarchical structure, which often make the concept relationships it presents unintuitive (for a discussion, see Gangemi et al., 2001).

## 3.3 Models

To ensure that the hyper-parameters are not biased towards the particular models we are introspecting, we use different models for selecting the hyper-parameters and the main experiments. We use GPT-2 (Radford et al., 2019) to select hyper-parameters discussed in Fig. 1 (e.g., the size of a positive and negative datasets) and validate that our data identifies a stable set of experts (Sec. 4.2). For all other experiments, we use models from the Pythia family (Biderman et al., 2023), specifically focusing on model sizes 70m (smallest), 1b, and 12b (largest), to understand the impact of model size on ExpertLens representations. For each model, we work with checkpoints 1, 512, 1k, 4k, 36k, 72k, and 143k, to track how ExpertLens representations develop throughout training. All Pythia models were trained on the same data presented in the same order, allowing us to evaluate the impact of model size and number of training steps on ExpertLens representations while controlling for the data/training recipe. Additionally, we show in App. E that our findings reported in the main text hold for more modern model architectures like Gemma-2b and Gemma-2b-instruct.

# 4 Results

## 4.1 Establishing the causal connection between expert units and model generations

We begin by replicating the causal connection between expert neurons and model generations found in prior work (Suau et al., 2023). Specifically, we find the expert neurons for the fifty concepts described in App. F.1 and activate top-500 (0.14 %) of them by setting them to their expected value over the positive set as described in Sec. 3.1 in Pythia-1b. We generate 5000 sentences each from the original and the intervened model with a neutral prompt "Once upon a time", following (Suau et al., 2023) (temperature = 1.0, max new tokens = 300; default parameters otherwise). We preprocess each generation by removing the prompt, tokenizing and lemmatizing the generated text using spaCy ('en_core_web_sm'). We keep only content words (nouns, verbs, adjectives, and adverbs) for the analysis.

To investigate the causal effect of intervention, we consider the prevalence of a list of words strongly associated with a given concept. To obtain this list, we prompt OpenAI GPT-4 to give a list of 50 words associated with the concept in question, and lemmatize these words in the same way as the model generations. To ensure the intervention is not simply boosting the exact words in the positive set, we remove any words that appear in the corresponding (lemmatized) documents in that concept's positive set, resulting in a list of between 8 and 43 *previously unseen* words per concept (median 23).

We find that intervening on a given concept leads to a significant difference in the prevalence of the corresponding concept-specific words (p<1e-5, evaluated using a two-sided permutation test). On average, there was a 0.181% increase in these previously unseen related words, with 58/60 concepts seeing an increase. Note, because we have excluded words that appear in the positive set, the remaining related words tend to be less common; if we do not exclude words appearing in the positive set, we see a 3.32% increase in prevalence of related words, with all concepts seeing an increase.

We provide sample generations for the concept 'table' from the original Pythia-1b model (with 0 experts activated) and from the intervened Pythia-1b model (with 500 experts activated) in App. B. As expected, we replicate prior work (Suau et al., 2023) – activating experts leads to the expression of the concept in model generations, again underscoring the well-established causal role of experts. We note here that activating experts does not necessarily lead to generating text that contains the target word itself but rather to generating text consistent with the concept expressed by the word (for example, activating the concept 'table' results in generations that can contain the word 'table' but also references to dining room, restaurant, eating with a family, etc.).

## 4.2 ExpertLens is stable across different dataset characteristics

If we intend to use ExpertLens for interpretability, it is important to establish that the identified neurons are robust to variation in the extraction procedure. To verify that this is the case, we conduct a pilot study to explore the impact of dataset size, the model used to generate the dataset, and the exact sentences used to represent a concept on the stability of the discovered expert sets.

For the pilot study, we sample 50 word pairs from the training split of the MEN dataset. For each concept in the word pair, we generate a positive set containing 7000 sentences from each of three models: GPT-4, Mistral-7b-Instruct-v0.2, and an internal 80b-chat model. We sweep over positive set sizes of 100, 200, 300, 400, and 500 sentences, and negative set sizes of 1000 and 2000 sentences. For each positive and negative set combination, we repeat expert extraction eight times (folds) with the sets randomly sampled from the full pool of sentences. We examine how sensitive the discovered experts are to the specific slice of the positive and negative sets (the 8 folds). We measure sensitivity in terms of the stability in experts across the folds, where high stability occurs when there is large overlap in the experts across folds. To assess overlap, we look at Jaccard similarity between expert sets across folds, using a range of thresholds $\tau$.

The findings are shown in Fig. 1 for each dataset configuration (subplot) and value of $\tau$ (x-axis). The expert neurons discovered across different data configurations and folds (indicated by the error bars) are stable, as indicated by a high ($\sim 0.8$ for $\tau = 0.5$) within-concept overlap proportion, and show little sensitivity to our manipulations. As $\tau$ increases, the overlap decreases, likely due to the shrinking expert set size (see Fig. 7).

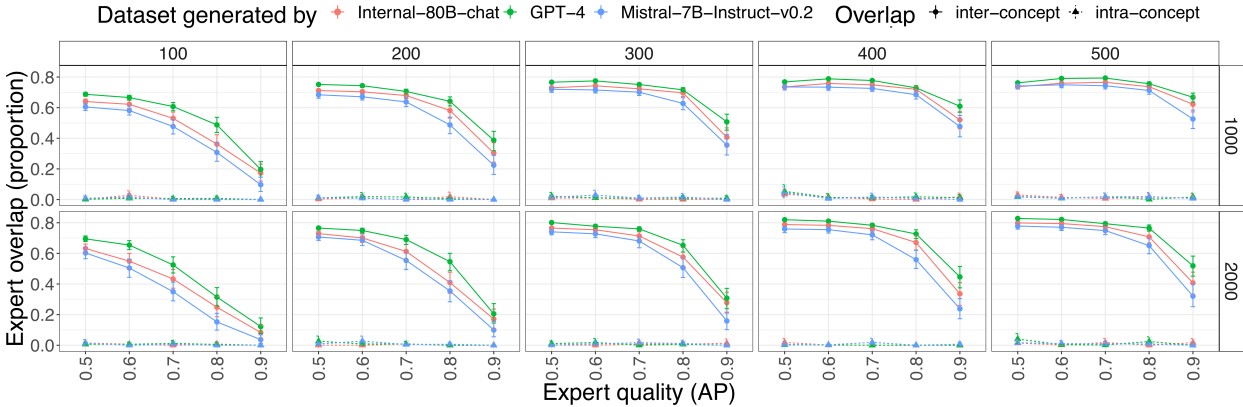

Figure 1: ExpertLens is relatively stable across various dataset characteristics. Points represent condition means; error bars represent bootstrapped 95% confidence intervals. Columns and rows represent the size (number of unique sentences) of the positive and negative sets respectively. Inter-concept is within-concept expert overlap; intra-concept is expert overlap averaged across randomly sampled pairs of concepts. See App. C for corresponding expert set sizes.

Conversely, the expert overlap for two different randomly sampled concepts is essentially 0 for all datasets and values of $\tau$. Taken together, this suggests that ExpertLens captures meaningful information about the target concept. Interestingly, the LLM (line color) used to generate the probing dataset matters little — while stronger models generate more diverse datasets (mean type/token ratio of 0.34, 0.21 and 0.18 for GPT-4, internal 80b-chat, and Mistral-7b-Instruct-v0.2 respectively), resulting in a somewhat higher expert overlap, the gain is too small to warrant their increased cost. Expert overlap increases with every increase in the size of the positive set, but the increases are small beyond 300 sentences, and performance for 400 sentences is virtually indistinguishable from 500 sentences. Interestingly, a larger negative set results in lower expert overlap at higher $\tau$ values and an increased variability across folds. One reason could be that as the size of the negative set increases so does the probability of the negative set containing sentences related to the target concept (e.g., a sentence about "cats" may also talk about "dogs"). A second explanation could be that larger negative sets activate more polysemous neurons.

Based on these findings, we conduct all subsequent analyses with a positive set of 400 sentences and a negative set of 1000 sentences, all generated with Mistral-7b-Instruct-v0.2.

### 4.3 ExpertLens representations are highly aligned with human representations

We now turn to the main question of our study — whether ExpertLens representations capture semantic information meaningful to humans. We assess this by measuring the alignment between expert-based and human representations. Specifically, for each pair of concepts, we look at the Jaccard similarity between expert sets for $\tau \in \{0.5, 0.6, 0.7, 0.8, 0.9\}$, taking this as an ExpertLens similarity score. In Fig. 2, we look at the correlation of these scores with human similarity measures from the MEN dataset, across various model checkpoints. We considered several more complex measures of expert-based similarity: cosine similarity between the raw AP values for two concepts and KL-divergence between the raw AP values for two concepts. We find similar correlations to those obtained with Jaccard similarity ($\tau = 0.5$), suggesting that what matters most is not the magnitude of the AP value, but rather whether it is above or below 0.5 (i.e., whether the neuron is positively or negatively associated with the concept). We focus on Jaccard similarity in the main text since it is significantly cheaper to calculate and present the cosine distance and KL-divergence findings in App. D.

**Expert neuron overlap is highly aligned with human similarity judgments** We find that ExpertLens representations are closely aligned with humans, with the highest alignment occurring at $\tau = 0.5$, Fig. 2. At

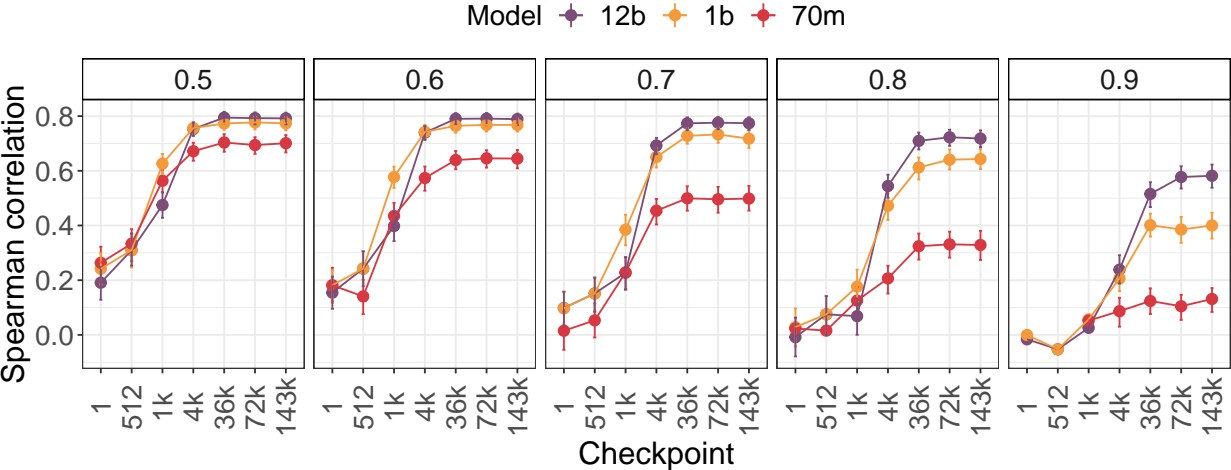

Figure 2: ExpertLens representations are closely aligned with human ones. Points are Spearman correlations between the expert neuron overlap and perceived human similarity in the MEN dataset (significant after checkpoint 1, p<0.05); error bars are bootstrapped 95% confidence intervals. The subplots are labeled with $\tau$.

the final checkpoint, the Spearman correlations between expert overlap ($\tau = 0.5$) and MEN similarity are 0.70, 0.77, and 0.79 for 70m, 1b, and 12b model respectively. For reference, agreement between humans has a correlation of 0.84. Interestingly, model size has only a small impact on this alignment (in line with findings in vision from Muttenthaler et al., 2023): ExpertLens representations in the 1b and 12b models are virtually indistinguishable, with the 70m model slightly less aligned. The models start diverging in how well aligned they are with humans as $\tau$ increases, with larger models being more aligned. This is because smaller models have fewer experts (Fig. 7) resulting in a lot of empty expert set intersections for higher levels of $\tau$.

To ensure that our findings generalize beyond the MEN dataset, we repeat our analysis on a subset of the Semantic Priming Project (SPP) (Hutchison et al., 2013), which contains $1,661$ target words paired with related or unrelated concepts. The advantage of the SPP dataset over MEN is that it contains a more varied set of concepts. The drawback is that the range of similarity levels between the concepts is more limited — SPP only contains three levels of similarity: strongly related, somewhat related, and unrelated concepts. We expect that expert overlap will increase as human-perceived similarity level increases. We sample 100 pairs from each of the three similarity bins in the SPP dataset and extract the experts for each concept in the pair from the final (143k) checkpoint of the three Pythia models under consideration. We then use linear mixed-effects regression to predict expert overlap from model (sliding difference coded[1]: 1b vs. 70m and 12b vs. 1b) and similarity level (sliding difference coded: weak vs. none and strong vs. weak). The model included the maximal converging random effects structure (random intercepts for the two concepts in a pair). For models of all sizes, we find a statistically significant increase in expert overlap with increased similarity (all p's $> 0.0001$; see Fig. 3).

**ExpertLens representations are more aligned than embeddings** Concept representations in the models have traditionally been captured through the analysis of model embeddings (Ettinger & Linzen, 2016; Auguste et al., 2017; Digutsch & Kosinski, 2023; Sajjad et al., 2022). We hypothesize that ExpertLens representations are more correlated with human representations than the embeddings as they better disambiguate different word senses. To test this, for each concept in the MEN test pair, we extract two types of embeddings from each model checkpoint: decontextualized single-word embeddings from the embedding layer in line with prior work on LLM-human concept alignment (Digutsch & Kosinski, 2023) and contextualized sentence embeddings (the average of the sentence embeddings from the positive set from the final hidden layer). We

---

[1]Sliding difference coding compares the mean of the dependent variable for one level of the categorical variable to the mean of the dependent variable for the preceding adjacent level (e.g., 1b model vs. 70m model).

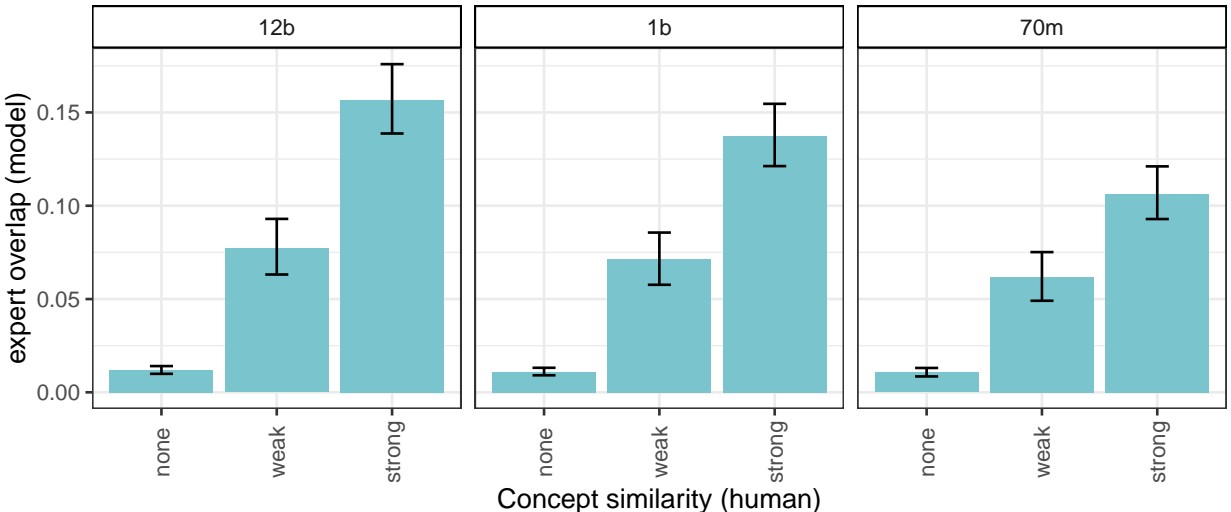

Figure 3: Expert overlap (Jaccard similarity) in the model is predicted by human-perceived similarity level in the SPP dataset. Bars represent expert overlap averaged over all concept pairs; error bars represent bootstrapped 95% confidence intervals. The subplots are model sizes.

compute cosine similarity between the embeddings for each word pair in the MEN test split as a measure of embedding similarity and correlate it with human similarity judgments.

We find that both contextualized and decontextualized embeddings are significantly correlated with human similarity judgments (p<0.05). However, when compared to the best-performing $\tau$ of Jaccard similarity (0.5), the correlations with human similarity are significantly lower for both types of embeddings compared to the experts (p-values<0.0001 and <0.05 comparing the alignment based on experts vs. single-word and sentence embeddings respectively, Fig. 4), supporting our hypothesis.

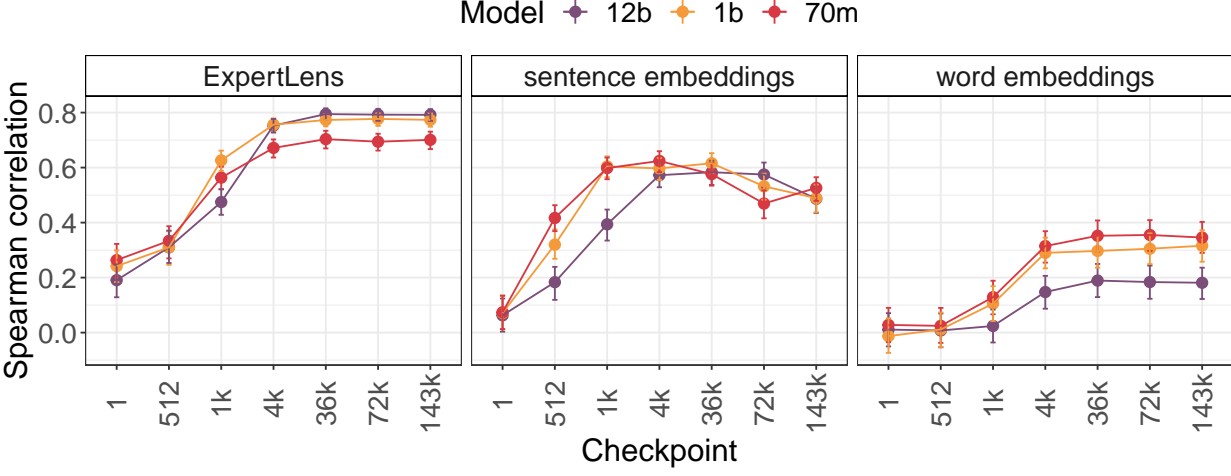

Figure 4: ExpertLens representations are more closely aligned with human ones than the embeddings. Points are Spearman correlations between LLM human similarities in the MEN dataset; error bars are bootstrapped 95% confidence intervals. Subplots are similarity type: ExpertLens are best-performing $\tau$ of Jaccard similarity (0.5), significant (p<0.05) after checkpoint 1; sentence embeddings are the average last-layer embeddings over the positive set, significant after checkpoint 1; single-word embeddings are from the embeddings layer, significant after checkpoint 4k for the 12b models and after checkpoint 1k for other sizes.

**ExpertLens representations are more aligned than sparse autoencoder features** We further compare ExpertLens representations to SAE features. To do so, we replicate our ExpertLens findings on Gemma-2-2b (Team, 2024). Additionally we use SAEs from the Gemma Scope suite (Lieberum et al., 2024) trained on the residual stream of Gemma-2-2b, with a dictionary size of 16k, to extract features for each concept of interest in the positive set. We pre-process the sentences as follows. We first lemmatize the sentences using spaCy ('en_core_web_sm') and create all morphological forms of the target concept with LemmInflect [2] (e.g., for the concept "abandon", we consider "abandon", "abandoning", "abandoned", etc.). For every morphological variant of the target concept, we extract SAE features at its token position from all layers of the model. We threshold the feature activations at 0.1 to remove noise. For our analysis, we consider both a) all features; and b) only features that occur in a proportion of at least $\tau$ of sentences associated with a context. The latter can be seen as analogous to the ExpertLens expertise threshold. For SAE features, we find that collapsing features across layers results in lower correlations with human similarity judgments than considering the best-layer correlations. We therefore present the best-layer correlations for each threshold in Fig. 5, which ends up being Layer 18-22 in practice for $\tau \geq 0.5$, and Layer 8 for $\tau = 0$ (i.e., no threshold).

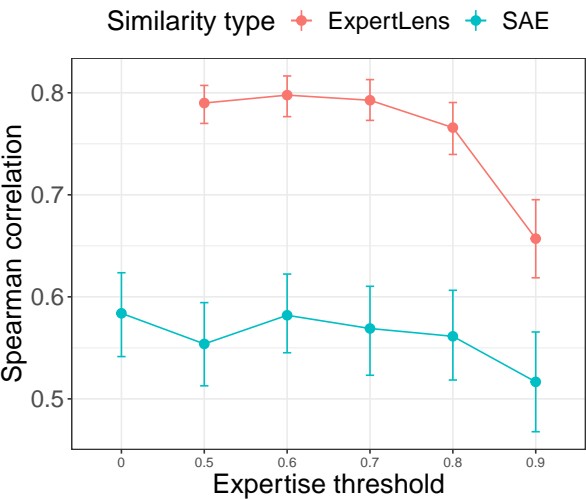

Figure 5: ExpertLens representations are more closely aligned with human judgments than best-layer SAE features in Gemma-2-2b. Points are Spearman correlations between LLM similarity and human similarity in the MEN dataset; error bars are bootstrapped 95% confidence intervals. For ExpertLens, expertise threshold is $\tau$; for SAE it is the percentage of sentences containing the feature.

We find that ExperLens representations are more aligned with human judgments than SAE features at all expertise thresholds of $\tau$ (Fig. 5). Our hypothesis is that due to the disentangled nature of SAE features, they are losing some of the concept associations captured by the ExpertLens, which matter for similarity. It is also possible that with a more thorough parameter sweep, SAEs would perform comparably to ExpertLens, which is essentially at the upper bound given that the human-human correlation is 0.84. ExpertLens will be preferred in that scenario as well since, unlike the SAEs, it does not require training.

**ExpertLens representations mirror human conceptual structure** Having established that the expert overlap is predictive of human-perceived concept similarity, we ask whether the ExpertLens captures a broader human-interpretable representation of concepts that goes beyond pairwise (dis)similarity. Specifically, we ask if the concepts are clustered in the expert space in a way that aligns with human-interpretable knowledge structures. Humans organize concepts into domains (Graf et al., 2016; Murphy, 2004; Rosch, 1978). For example, "dog", "cat" and "horse" are all *animals* and "bike", "bus", and "car" are all *vehicles*. This raises the question of whether we can reconstruct this type of organization from ExpertLens representations. To assess this, we consider a list of fifty concepts organized into ten domains (Sec. 3.2 and App. F.1), the experts associated with each concept in the list ($\tau$=0.5), and their Jaccard similarity. For this analysis, we consider only the final (143k) checkpoint. We discuss Pythia 12b in the main text and present other model sizes in App. F.2.

Fig. 6 provides a visualization of the concept structure in the expert space, revealing a clear domain organization: concepts belonging to the same domain are strongly associated (e.g., all color terms are connected to each other, but not to other domains), while cross-domain associations are notably sparser. On top of that, Fig. 6 shows meaningful between-domain connections unveiled through ExpertLens. For instance, while "driver" is an *occupation*, its expert set is also strongly associated with "bus" or "vehicle". Similarly, "racing" connects the *sports* domain with the *vehicles* domain. Finally, looking at the internal organization

---

[2]https://github.com/bjascob/LemmInflect

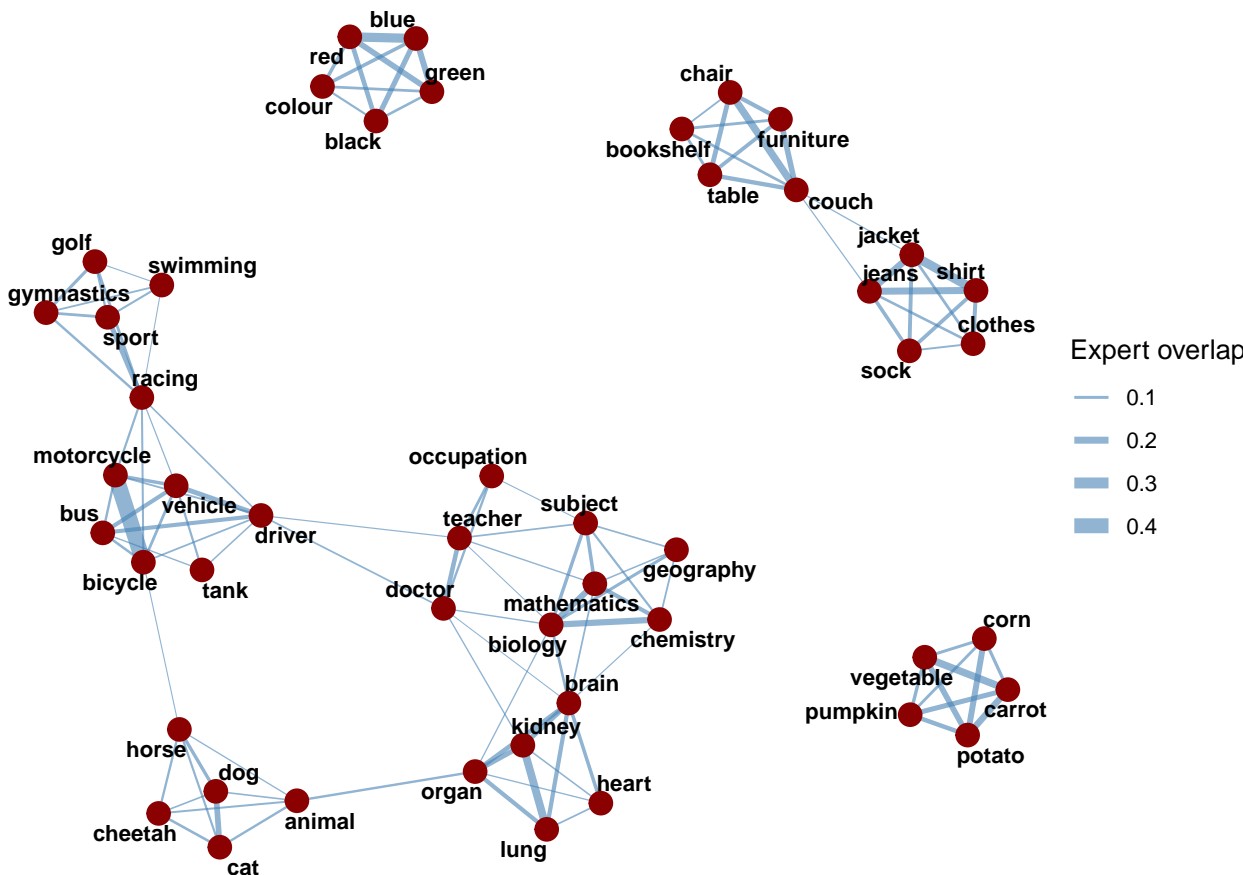

Figure 6: ExpertLens representations reconstruct human conceptual structure in Pythia-12b. Each node represents a concept; edge thickness corresponds to Jaccard similarity between concepts in the expert space.

of the domains, we notice that broader concepts (e.g., "vehicle" or "animal") tend to show weaker overlap with specific instances in their domain compared to the overlap between closely related specific concepts, e.g., "motorcycle" and "bicycle", or "dog" and "cat". This may reflect distributional factors, with narrower concepts exhibiting stronger co-occurrence patterns.

To further quantify whether domain structures emerge in ExpertLens representation, we test whether concepts from the same domain (e.g., "dog", "cat", "horse", and "cheetah") share a consistent set of experts, and whether some of these shared experts are also associated with the broader concept describing the domain (e.g., "animal" in our example). Our results reveal a clear and systematic pattern: within each domain, a consistent set of expert neurons is shared across all associated concepts. On average, 2.24% of the experts identified across all concepts in a domain are jointly shared among them. Notably, 58.45% of this shared core is also shared by the broader concept representing the domain (see App. F.2 for the complete result set). To validate the significance of our findings, we compare them against a baseline in which domain groupings are randomly sampled (e.g., associating "animal" with "jacket", "liver", "doctor", and "red"). In this case, the overlap among expert sets drops significantly (average 0.01% and 5.81% of shared neurons for all concepts and by the broader concept respectively, p-values <0.001) confirming that the structure we observe is unlikely due to chance.

Overall, our findings suggest that ExpertLens representations capture human-interpretable domain-level structures beyond simple word pair similarity.

## 4.4 Characterizing the discovered experts

We now consider how and where experts arise within the model, exploring differences in the expert sets discovered in Sec. 4.3 across model size and stage of training.

**Experts are learned from the data, with larger models having more experts**  Larger models allocate more experts to a given concept (see Fig. 7; the pattern does not change after scaling the raw number of experts by the number of neurons in the model, Fig. 11). As $\tau$ increases and experts become more specialized, fewer experts are identified; the drop is more pronounced for smaller models. Overall, larger models have a greater capacity to learn a higher number of experts and a higher number of *more specialized* experts. This increased specialization may contribute to finer-grained concept representations and ultimately better performance on downstream tasks.

Interestingly, we observe a large number of experts at checkpoint 1, followed by a drop and then a steady gradual increase in the number of experts as training continues. This could be explained from the perspective of language modeling as compression (Shwartz-Ziv & Tishby, 2017; Delétang et al., 2024). Early in training, the model discovers a large number of experts. While they are not yet meaningful (as indicated by non-significant correlation with human similarity judgments), they allow the model to efficiently allocate representational capacity for later in training. As the model starts learning the relevant relationships, the number of experts drops (checkpoint 512) and then slowly recovers as the model continues learning (checkpoint 1k onwards). As training continues, the experts become more meaningful, as evidenced by the increasing correlation between the expert overlap and human similarity judgments. The idea that experts are learned from training data is further supported by the fact that we find a mode of 0 experts in all models initialized with random weights.

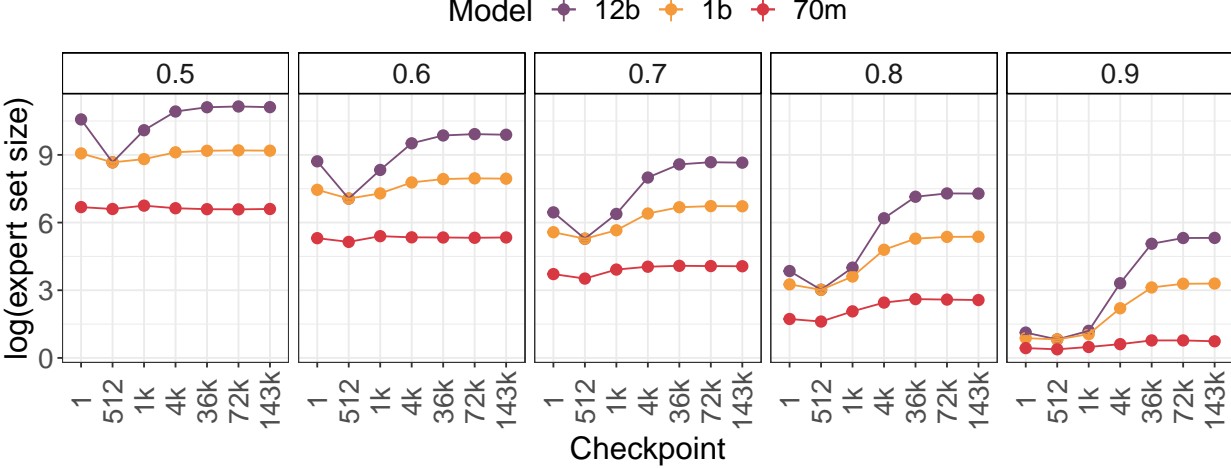

Figure 7: Expert set size (log) by model size and checkpoint. Points are averages over all concepts; error bars are bootstrapped 95% confidence intervals. Subplots are different values of $\tau$.

**More specialized experts take longer to learn**  We next look at the dynamics of learning experts across checkpoints. We calculate expert overlap (Jaccard similarity) for each concept across subsequent checkpoints in our data. The stability of the discovered expert set grows as training progresses (Fig. 8). Early in training (prior to step 36k), expert overlap between subsequent checkpoints is low across model sizes, suggesting that semantic knowledge has not been acquired yet. As $\tau$ increases (corresponding to higher expert specialization), it takes longer for the expert set to stabilize, suggesting that higher-quality experts take longer to learn.

**Expert location varies with expertise level**  Overall, we find more experts in the MLP compared to attention layers in models of all sizes (after controlling for the number of neurons in the respective layers), with the relative allocations stabilizing at checkpoint 4k, App. G.1. Interestingly, the distribution of expert neurons across layers changes depending on the value of $\tau$. As $\tau$ increases and the expert set becomes more

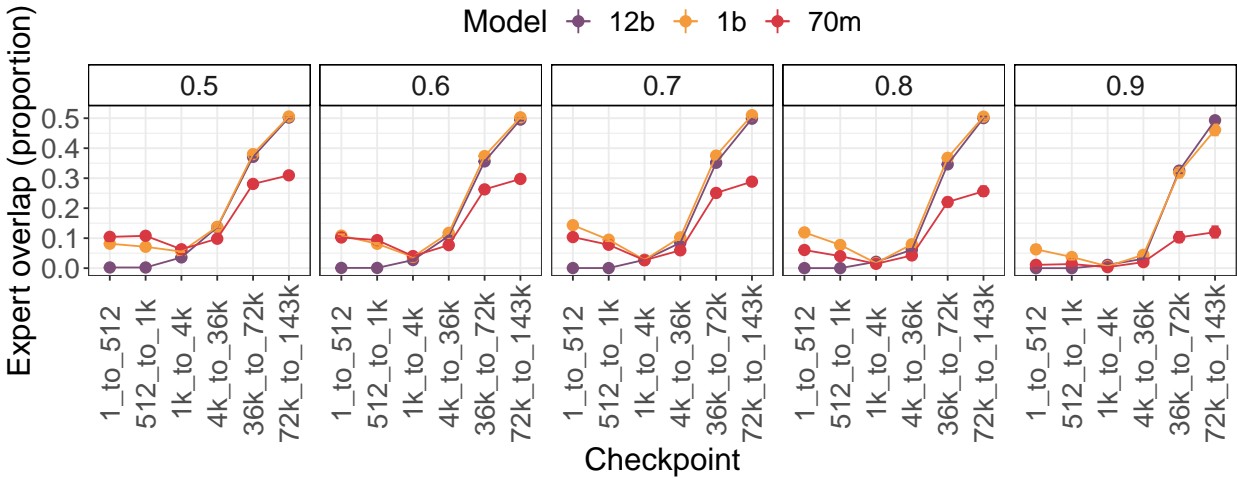

Figure 8: Expert overlap across subsequent checkpoints (e.g., 1_to_512 is overlap between checkpoints 1 and 512). Points are across concept averages; error bars are bootstrapped 95% confidence intervals. Subplots are different values of $\tau$.

predictive of the concept, the prevalence of experts gradually shifts from later layers to earlier layers in the MLP, while in the attention layers, the pattern shifts from being roughly uniform to bimodal (peaking in the middle and early layers), see App. G.2 and App. G.3. It is possible that different levels of $\tau$ reflect different aspects of the input captured by the units — for instance, some experts might capture surface-level characteristics of the concept while others capture the semantics. We find, however, no difference in the distribution of AP-values between the units shared by the two concepts in a pair vs. units privileged to each of the concepts (Fig. 9). Similarly, we find no difference in the location of the experts for concepts with broader vs. narrower meanings (e.g., "animal" vs. "dog"), see App. G.4 and App. G.5.

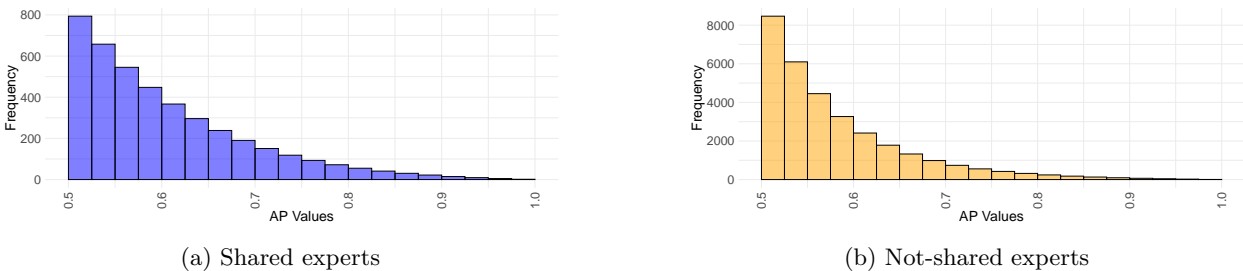

(a) Shared experts             (b) Not-shared experts

Figure 9: Histograms of raw AP values for the experts shared (blue) and not shared (yellow) between the concepts in a pair in Pythia-12b at checkpoint 143k (see App. H for other model sizes).

## 5 Discussion

Our work shows that concept representations captured with ExpertLens are stable across models and datasets and are closely aligned with humans, which underscores the suitability of ExpertLens as a tool for model interpretability. Coupled with the fact that ExpertLens is lightweight and data efficient, it opens a new avenue for interpretability at scale.

In this work, neurons are studied individually. That is, our analysis assumes that the representation of concepts is aligned with the canonical basis induced by the neurons. We have two reasons to assume that this is the case. First, we replicate prior work showing that intervening on expert neurons steers generations to express the concept (Suau et al., 2023; 2024). Second, in our analysis we see that neurons identified in this

manner capture key properties of concepts: the correlation between ExpertLens concept similarity measures and human concept similarity evaluations is comparable to inter-human correlation. It is, however, possible that looking at neurons jointly would capture additional aspects of concept representation. We leave this exploration to future work.

Prior work on transformer models has suggested that their neurons tend to be polysemantic — i.e., they activate for multiple concepts (Elhage et al., 2022; Scherlis et al., 2025). Our findings relate to this work. Each individual neuron in the expert set is indeed polysemantic — we see for example that neurons in the expert set for 'cat' tend to be in the expert set for "dog". However, we find that expert neurons are polysemantic along human-interpretable lines — experts for "cat" do not tend to be in the expert set for "car". There are, of course, neurons that activate for multiple unrelated concepts. However, these neurons are not predictive of those concepts and so do not appear in the expert set. Moreover, we have reason to believe that the predictive neurons (i.e., the experts) are the key drivers of concept-based behavior: Intervening on the predictive neurons (i.e., experts) increases the probability of the concept being expressed in the generations, while intervening on non-expert neurons does not (Sundar et al., 2025).

In this work, we assess the interpretability of expert-based representations based on single-word everyday concepts since we have the best measures of human-perceived similarity for these concepts. We find that the expert sets discovered for these concepts are stable across datasets and models and that model size does not play a significant role in expert discovery: we find similar patterns in experts for 12b and 70m in our setup. While this finding is consistent with previous literature (Muttenthaler et al., 2023) and replicated over two datasets, it is also possible that our task is too simple to distinguish between the models. This is supported by the observations that semantic relationships studied here start emerging early in training (around checkpoint 4k out of 143k).

Future work will consider more complex concepts such as those expressing human values or preferences (e.g., "toxicity" or "helpfulness"). We see potential uses for this approach as a tool for studying representational alignment in a variety of domains. Given our definition of a concept as a set of examples, it can be readily extended to more abstract concepts like safety, toxicity or value alignment. For instance, in safety alignment, one could ask questions such as: do existing alignment methods truly make the representations more aligned? There is an 'alignment tax' associated with alignment meaning that, after applying safety alignment, model performance drops on other tasks (Askell et al., 2021). Is this because other aspects of the representation become misaligned? Understanding these questions could lead to improved alignment, while providing insight into how to mitigate the undesirable consequences of applying changes to model representation. Going beyond alignment, ExpertLens could be a promising tool to study the relationship between the training data and knowledge representation in the model, which could guide us to design better synthetic datasets.

We hope that this work will serve as a foundation for future research not only in machine learning, but also at the intersection of cognitive science and AI theory, exploring whether fundamental cognitive principles (Murphy, 2004; Margolis & Laurence, 2003) are reflected in neural network representations.

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

## A    Prompts used for probing dataset generation and sample generations

---
**Fact prompt**

Generate a set of 10 sentences, including as many facts as possible, about the concept [concept name] as [a/an] [adjective/noun/verb] and defined as [WordNet definition]. Refer to the concept only as [concept name] without including specific classes, types, or names of [concept name]. Make sure the sentences are diverse and do not repeat.

---

**Sample fact sentences**    for concept **poppy** defined as "annual or biennial or perennial herbs having showy flowers":

---
**GPT-4**

Gardeners often classify poppies as easy to care for due to their hardy nature.

---

**Mistral-7b-Instruct-v0.2**

Poppies are herbaceous plants that can grow annually, biennially, or perennially, depending on the specific species.

---

**Internal 80b-chat model**

Poppies have been used in traditional medicine for centuries, with various parts of the plant being employed to treat ailments like pain, insomnia, and digestive problems.

---

**Story prompt**

Generate a set of 10 sentences, where each sentence is a short story about the concept [concept name] as [a/an] [adjective/noun/verb] and defined as [WordNet definition]. Refer to the concept only as [concept name] without including specific classes, types, or names of [concept name]. Make sure the sentences are diverse and do not repeat.

---

**Sample story sentences**    for concept **poppy** defined as "annual or biennial or perennial herbs having showy flowers":

---
**GPT-4**

As the wedding gift from her grandmother, a dried poppy was framed and hung on her wall.

---

**Mistral-7b-Instruct-v0.2**

As the farmer tended to his fields, he couldn't help but admire the poppies that grew among his crops, their beauty a welcome distraction.

---

**Internal 80b-chat model**

The poppy, a harbinger of spring, adorned the hillsides with a colorful tapestry, signaling the end of winter's slumber.

---

## B    Sample generations for the concept "table"

The model (Pythia-1b) is prompted with a neutral prompt "Once upon a time". Words and phrases associated with the concept "table" have been manually annotated. For the intervened model, we provide two sample generations to illustrate the variability.

**Concept 'table' original model**

Once upon a time, this area of the ocean was covered by a thick mist that gave the area the name "The Mist". Since then, many new species of marine invertebrates like sand wriggler and whelk have evolved and multiplied. The first people to walk upon or explore the ocean's deep layers were the Chinese. They named the region, "The Sea Wall", and the Chinese have continued to explore the ocean bottom to this very day, following in the footsteps of ancient explorers and scientists. What we find today can be traced back hundreds of millions of years. You may wonder what has attracted these organisms to land. To most of us, it appears the ocean is a world of mystery and wonders. To the marine biologists, this means they have the gift of seeing things beyond our eyes. They are able to look deep up in the ocean, below the surface of the sea, to far away parts of the ocean they have barely heard of before. A typical whale is about the size of a car, and is about the weight of a brick. The whale uses its massive front fins to propel itself forward and to the sides, creating powerful propulsive thrust for a powerful body and a strong blow. As the whale advances forward, it has to adjust it body shape as they pass, so it is longer at the front, its body thickened. While many animals dive into the sea to hunt, others swim underwater to mate.

**Concept 'table' 500 experts activated**

Once upon a time everyone seemed to be so generous: the people in town were so generous, the neighbors were so kind. But since our home was so old, our generosity was a very small bit of the family's income. And so every night when we arrived at the restaurant from work, I would bring a few items from home, and the waitress would bring me what my family had left on the sideboard, and we'd eat with the family. The next morning when everyone was gone, I'd clean up and go over the dining room. I was so happy I could have laughed for years afterwards to remember our first evening of working together. But even before that time, I'd find an unused spot in the dining room for my collection, just sitting there. I'd keep it by the door, where there was already a glass, and the place would be cleared and tidied up, no one being the least bit bothered, so I'd go downstairs and grab a few things, or my collection would be just fine. Sometimes it worked, sometimes it wasn't the case, but I'd pick up all my belongings when I had to. There aren't many things that have brought so much joy to my life, that I'm glad one bit of it was gone, since it was the only time it ever happened. However, one of the things I miss most is how I can just go visit family now and again.

**Concept 'table' 500 experts activated**

Once upon a time, for the sake of speed, a new piece may be added to the front of the stage, not yet visible, that will be revealed as the show progresses. There is no need to have a backstage area. In the past, with many large theatrical productions, if the actors have their backs to the audience and the front of the stage is visible to them, they cannot see the stage as well. It is a significant loss of audience attention and thus the audience may get impatient as well, causing the show to slow down. Another difficulty with having the stage in this setting was that the audience was faced with two separate sets, as the audience's seat is positioned at the rear of the stage, as the audience is not required to get up to view the stage. For the past 20 years or so, it has been standard practice to use a stage which can be turned into a bed or seat and thus not have a table adjacent to the audience. The present inventor has recognized the need to use a table in this manner for many other reasons. For instance, when using the stage in large theatrical productions, a table should be positioned on the stage in such a manner that when the actors have their backs to you, as was the case historically, it is much easier, if not mandatory, for them to see the stage and so the audience cannot become impatient or upset. Also with using tables for long periods of time, the space between seats becomes very narrow.

## C   Expert set sizes

We find that the number of experts for a given threshold $\tau$ decreases approximately logarithmically with $\tau$ (Fig. 7). This finding is consistent across models and positive/negative set sizes.

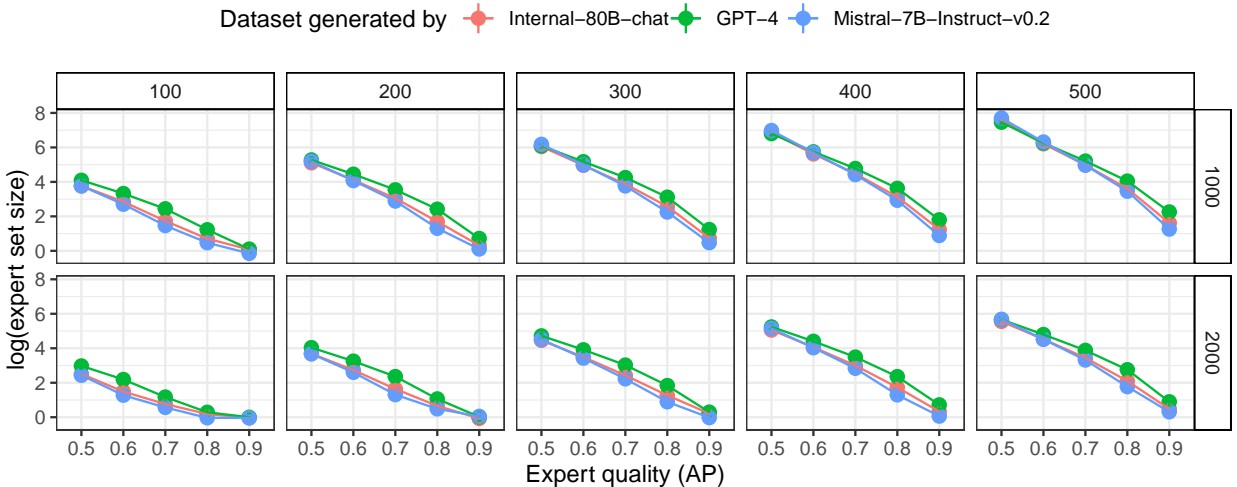

Figure 10: Expert set size (log) in the pilot experiment. Points represent condition means; error bars represent bootstrapped 95% confidence intervals. Columns represent the size of the positive set (number of unique sentences); rows represent the size of the negative set (number of unique sentences).

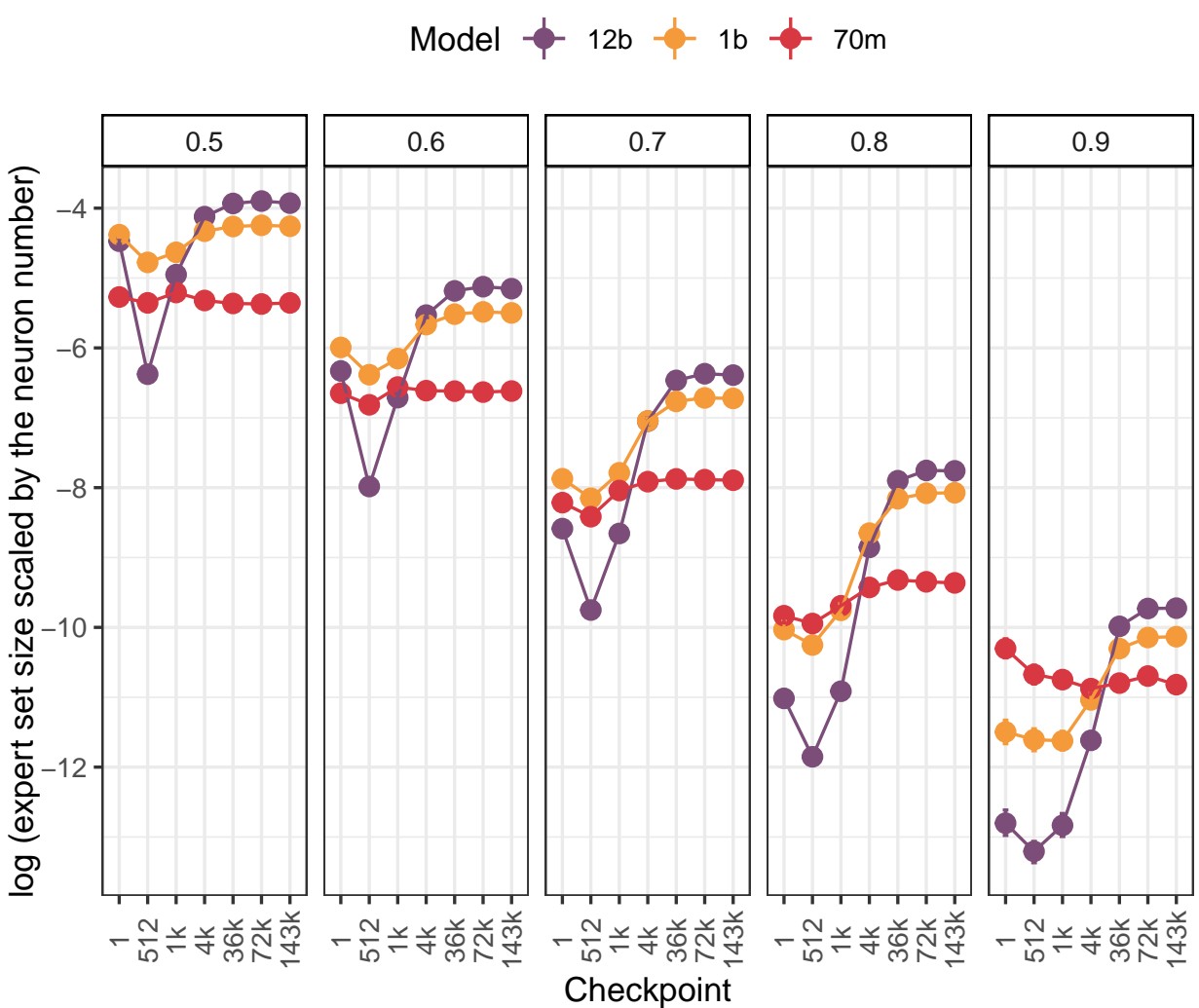

Figure 11: Expert set size (log) scaled by the number of neurons in the model in the main experiments on the MEN dataset. Points are averages over all concepts; error bars are bootstrapped 95% confidence intervals. Subplots are different values of $\tau$.

# D   Analyses of correlations between human similarity judgments and threshold-free metrics (cosine similarity and KL divergence)

In Sec. 4.3, we used Jaccard similarity to measure similarity between expert sets. Here, we look at alternative measures of similarity that do not require an expertise threshold $\tau$. In Fig. 12, we use cosine similarity over raw AP values, and in Fig. 13, we use symmetrized KL divergence. In both cases, we see a similar pattern to that seen for Jaccard similarity (Fig. 2). Note, since KL is a divergence rather than a similarity measure, the correlations are negative.

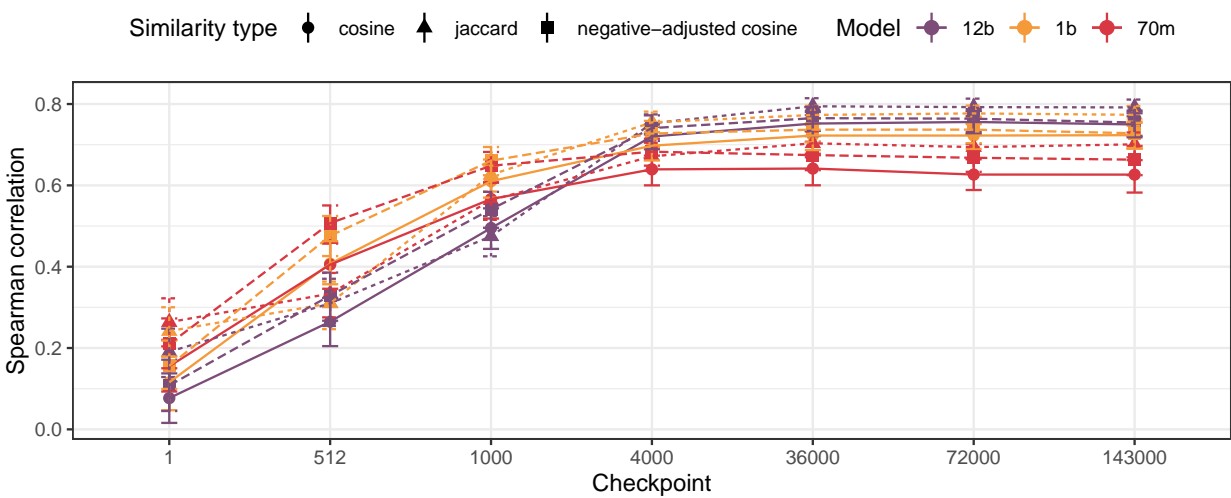

Figure 12: Spearman correlations between human similarity judgments, cosine similarity over raw AP values, negative-adjusted cosine similarity [abs(AP)-0.5], and the best-performing $\tau$ of Jaccard similarity (0.5). Points represent Spearman correlations between cosine similarity and perceived human similarity in the MEN dataset; error bars represent bootstrapped 95% confidence intervals.

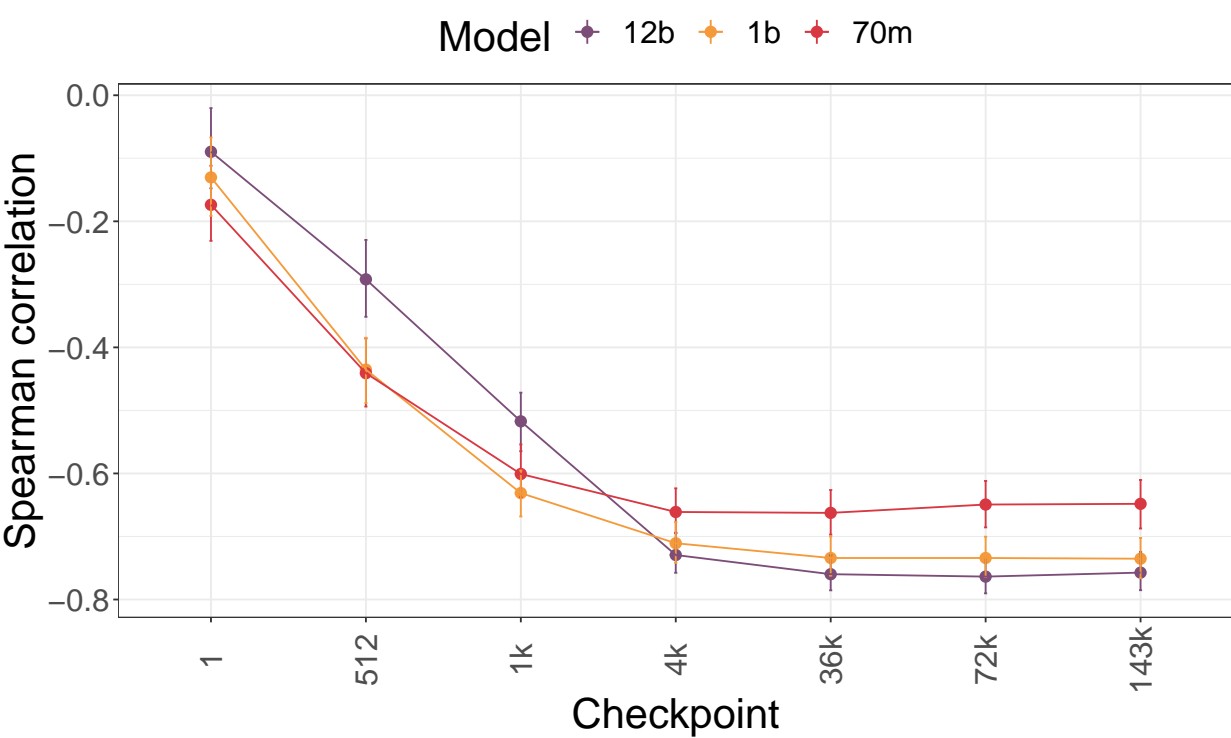

Figure 13: Spearman correlations between human similarity judgments and symmetrized KL divergence $D_{\mathrm{KL}}(c1 \parallel c2) + D_{\mathrm{KL}}(c2 \parallel c1)$ over raw AP values. Points represent Spearman correlations between KL divergence and perceived human similarity in the MEN dataset; error bars represent bootstrapped 95% confidence intervals.

## E    Correlations between expert representations and human similarity in Gemma-2b

Our experiments in the main text focus on the Pythia family of models, since we study ExpertLens development over training in different model sizes while controlling for training data/regime. Since Pythia models are not instruction tuned, in this section we turn to a different model family to look at how instruction tuning impacts the alignment of ExpertLens representations. Fig. 14 shows the correlation between expert neuron similarity and human similarity judgments in the MEN dataset for the pretrained Gemma-2b model, and the instruction-tuned Gemma-2b-it model. We see that the instruction-tuned model has, on average, slightly higher correlation with human perceived similarity; however, the difference is not significant. Overall, the Gemma models show comparable alignment to similarly-sized Pythia models.

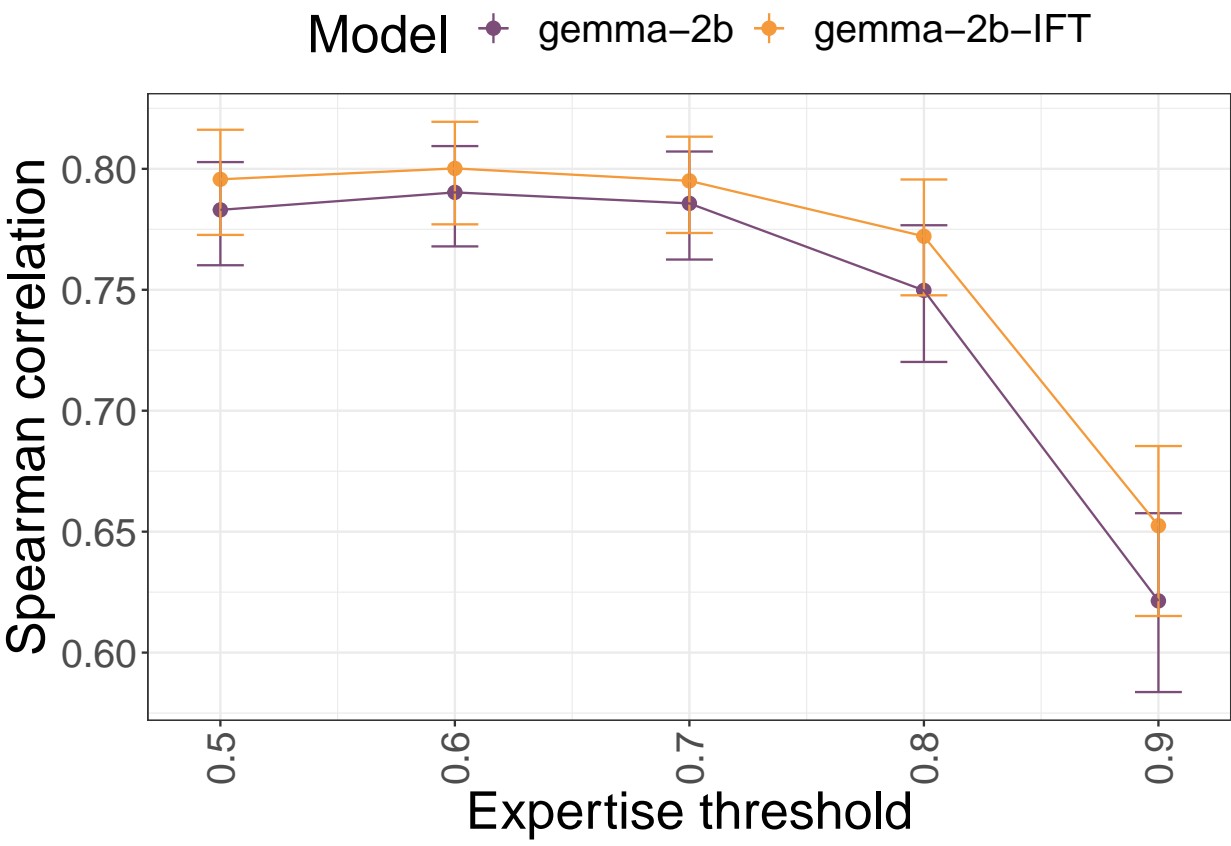

Figure 14: ExpertLens representations in Gemma-2b and Gemma-2b-it models are closely aligned with human ones. Points are Spearman correlations between the expert neuron overlap and perceived human similarity in the MEN dataset (all statistically significant, p<0.0001); error bars are bootstrapped 95% confidence intervals.

# F Domain-based analyses

## F.1 List of concepts in semantically-related domains

Table 1 provides a list of concepts used for studying concept organization in Sec. 4.3 and the causal role of expert neurons in model generations App. B. This list was manually curated by the authors.

| Domain | Concepts |
|--------|----------|
| animals | cat, dog, cheetah, horse, animal |
| clothes | jacket, jeans, shirt, sock, clothes |
| colours | red, blue, green, black, colour |
| furniture | chair, bookshelf, table, couch, furniture |
| occupations | doctor, teacher, driver, musician, occupation |
| organs | heart, kidney, lung, brain, organ |
| sports | golf, racing, gymnastics, swimming, sport |
| subjects | mathematics, geography, biology, chemistry, subjects |
| vegetables | carrot, potato, pumpkin, corn, vegetable |
| vehicles | bus, tank, motorcycle, bicycle, vehicle |

Table 1: List of concepts in our domains.

## F.2 Domain-level organization results for Pythia-70m and Pythia-1b

Fig. 6 in Sec. 4.3 shows that ExpertLens representations can reconstruct human-interpretable concept domains in the Pythia-12b model. Fig. 15 and Fig. 16 show this reconstruction for the Pythia-70m and Pythia-1b models respectively. Table 2 provides the baseline and statistical significance testing for neuron overlap discussed in Sec. 4.3 for models of all sizes under consideration.

| Model | Ckpt | % experts shared in domain | % experts shared with broader concept |
|---|---|---|---|
| 70m | 1 | 0.05 0.00 | 0.00 0.00 |
| | 36k | 0.97 0.01 | 70.41 0.33 |
| | 72k | 1.19 0.00 | 59.69 0.33 |
| | 143k | 1.39 0.01 | 67.19 0.92 |
| 1b | 1 | 0.02 0.00 | 0.24 0.33 |
| | 36k | 1.81 0.01 | 60.87 2.67 |
| | 72k | 1.84 0.03 | 63.43 2.24 |
| | 143k | 2.02 0.01 | 63.84 2.85 |
| 12b | 1 | 0.12 0.00 | 0.01 0.52 |
| | 36k | 1.87 0.01 | 58.66 5.11 |
| | 72k | 2.12 0.01 | 57.85 5.50 |
| | 143k | 2.24 0.01 | 58.45 5.81 |

Table 2: Results of expert overlap in semantically-organized domains, across different models and checkpoints. *% shared in domain* shows the average percentage of experts shared between all the specific concepts in a domain (e.g., "dog", "cat", etc. ). Column 4 reports the percentage of this shared core also activated by the broader concept representing the domain (e.g., "animal"). Baseline values are shown in gray. Our results are significantly different from the randomized baseline starting from checkpoint 36k, suggesting that domain-like structures seem to have fully emerged at that stage.

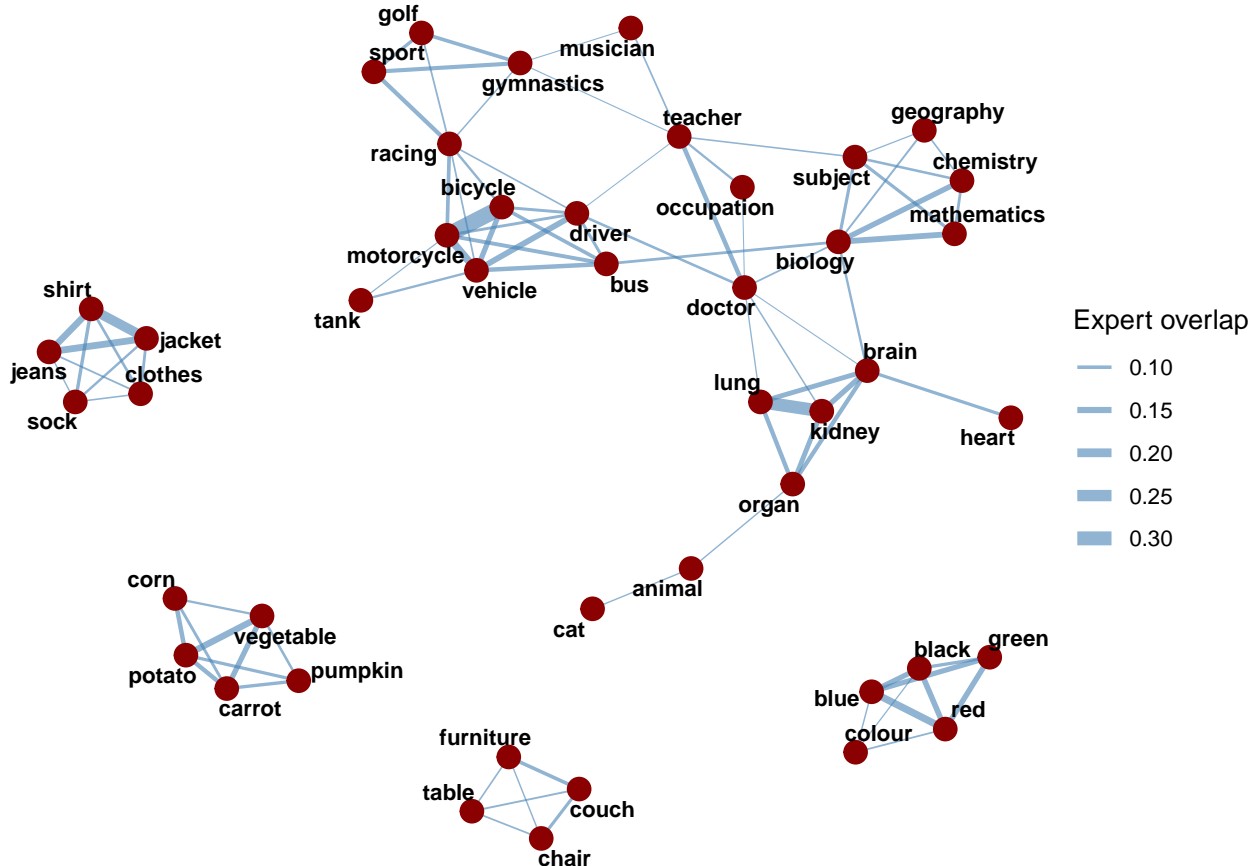

Figure 15: **Pythia70m, ckpt 143k**ExpertLens representations reconstruct human conceptual structure. Each node represents a concept; edge thickness corresponds to Jaccard similarity between concepts in the expert space.

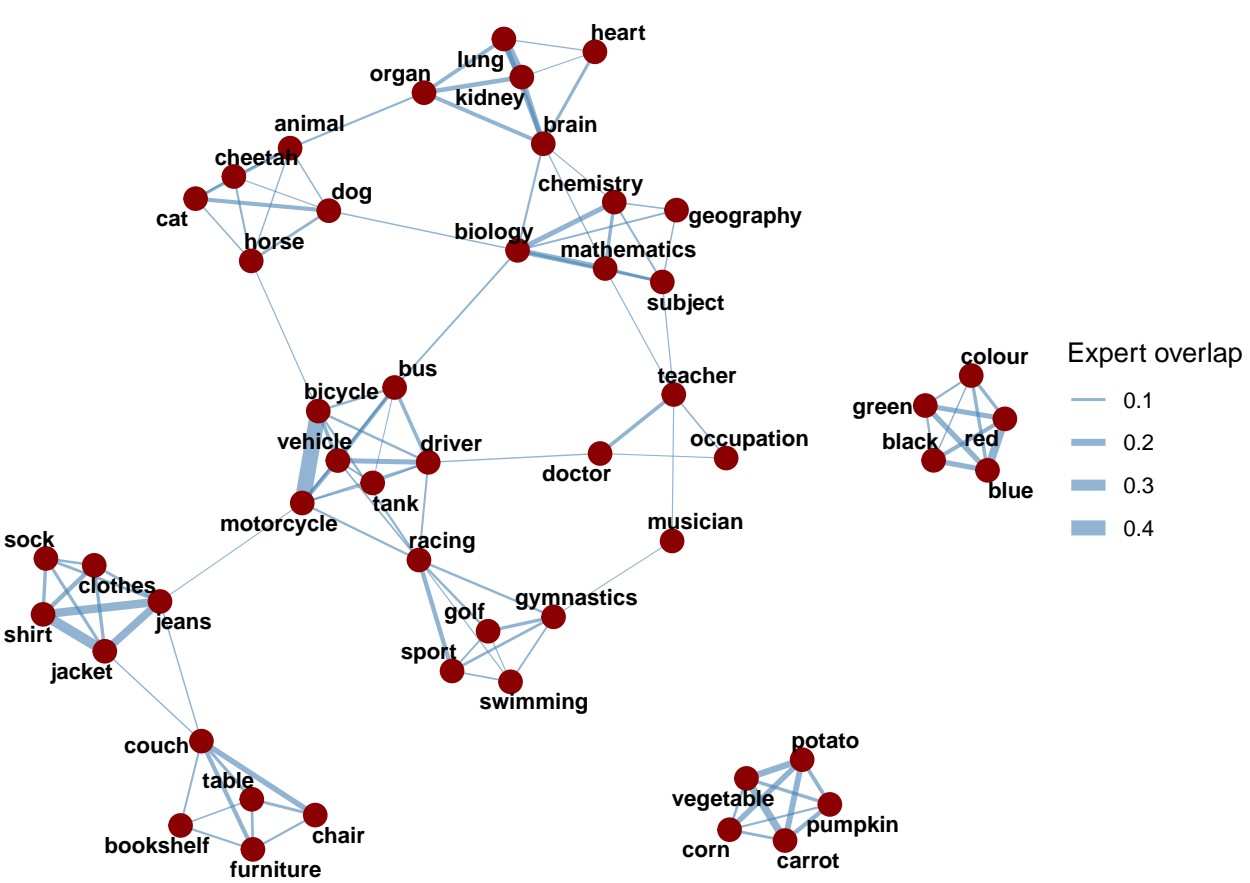

Figure 16: **Pythia1b, ckpt 143k** ExpertLens representations reconstruct human conceptual structure. Each node represents a concept; edge thickness corresponds to Jaccard similarity between concepts in the expert space.

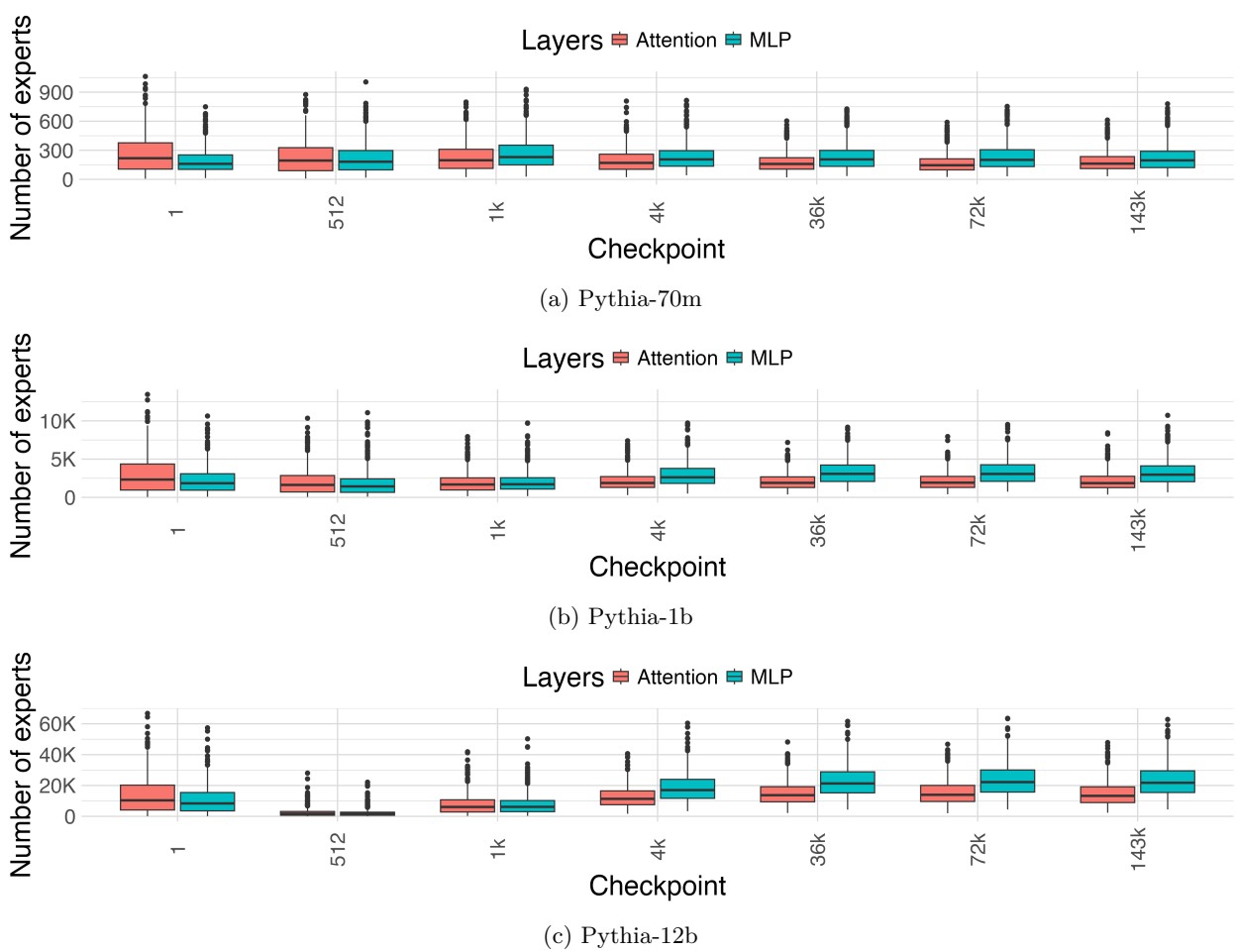

Figure 17: The distribution of experts across the attention and MLP layers in the 70m (top), 1b (middle), and 12b (bottom) Pythia models. Attention layers are shown in pink; MLP layers are shown in blue.

## G    The distribution of expert neurons in the network

In this section, we look at where in the network the discovered expert units are located.

### G.1    The distribution of expert neurons in the attention and MLP layers

Fig. 17 shows the distribution of experts across MLP and attention layers for the three Pythia model sizes under consideration. We find overall more experts in the MLP layers.

### G.2    The distribution of expert neurons in the MLP layers as a function of $\tau$ value

We find that the distribution of expert neurons in the MLP varies based on the $\tau$ value in models of all sizes. Fig. 18,  19, and  20 show the distribution of experts in the MLP layers for the five $\tau$ thresholds considered in this work for Pythia-70m, Pythia-1b, and Pythia-12b models respectively.

### G.3    The distribution of expert neurons in the attention layers as a function of $\tau$ value

We find that the distribution of expert neurons in the MLP varies based on the $\tau$ value in models of all sizes. Fig. 21,  22, and  23 show the distribution of experts in the attention layers for the five $\tau$ thresholds considered in this work for Pythia-70m, Pythia-1b, and Pythia-12b models respectively.

### G.4 Distribution of experts for broader and narrower concepts in the MLP layers

We look at the distribution of expert neurons across MLP for the broader vs. narrower concept (e.g., "animal" vs. "dog") for the concepts in App. F.1. We find no difference in their distribution in the MLP layers (see Figures 24, 25, 26 for Pythia-70m, Pythia-1b, and Pythia-12b respectively.

### G.5 Distribution of experts for broader and narrower concepts in the attention layers

We look at the distribution of expert neurons across attention for the broader vs. narrower concept (e.g., "animal" vs. "dog") for the concepts in App. F.1. Similarly, to the findings for the MLP layers (App. G.4), we find no difference in expert distribution in the attention layers (see Figures 27, 28, 29 for Pythia-70m, Pythia-1b, and Pythia-12b respectively.

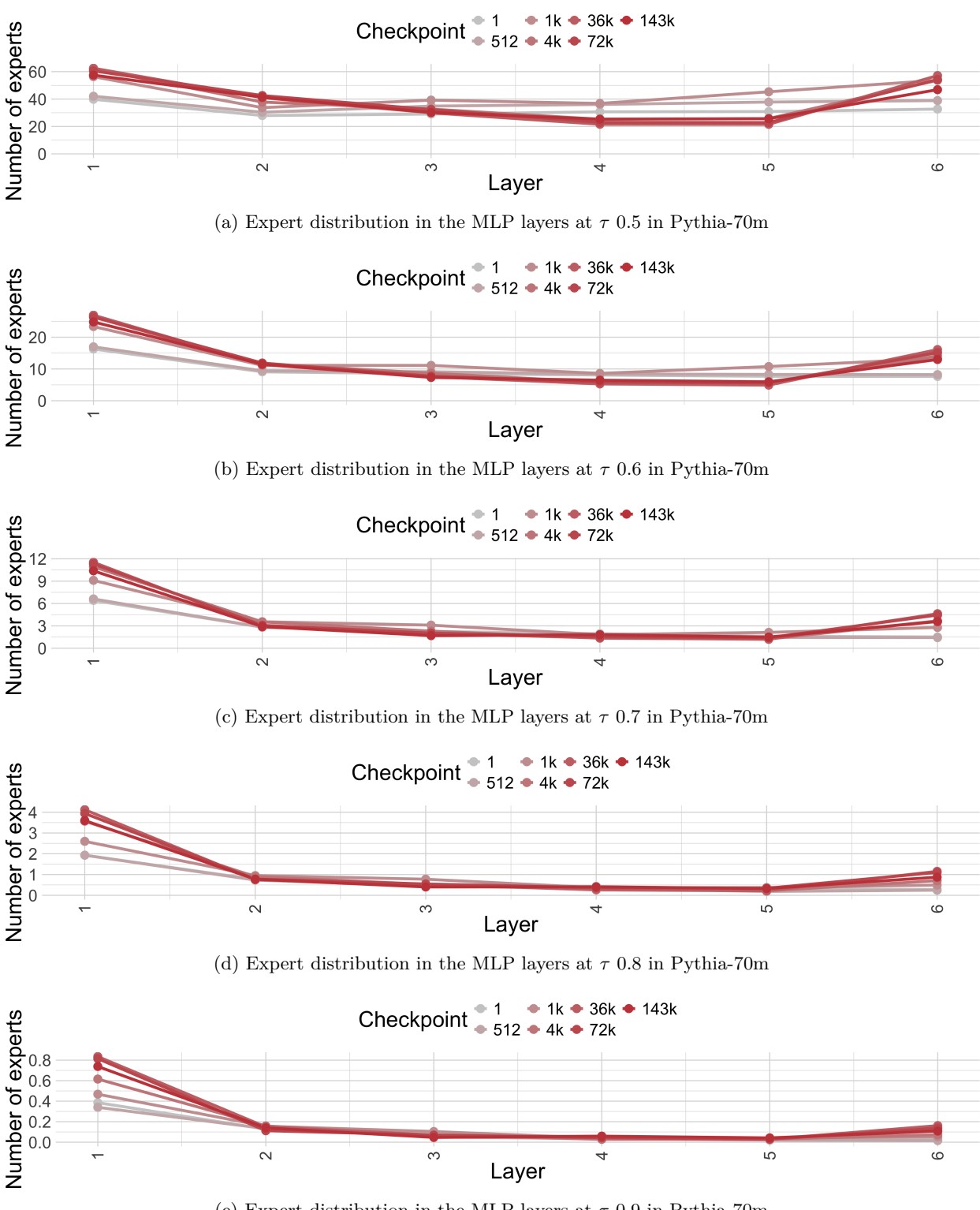

(a) Expert distribution in the MLP layers at $\tau$ 0.5 in Pythia-70m

(b) Expert distribution in the MLP layers at $\tau$ 0.6 in Pythia-70m

(c) Expert distribution in the MLP layers at $\tau$ 0.7 in Pythia-70m

(d) Expert distribution in the MLP layers at $\tau$ 0.8 in Pythia-70m

(e) Expert distribution in the MLP layers at $\tau$ 0.9 in Pythia-70m

Figure 18: The distribution of experts across in MLP layers in the Pythia-70m as a function $\tau$.

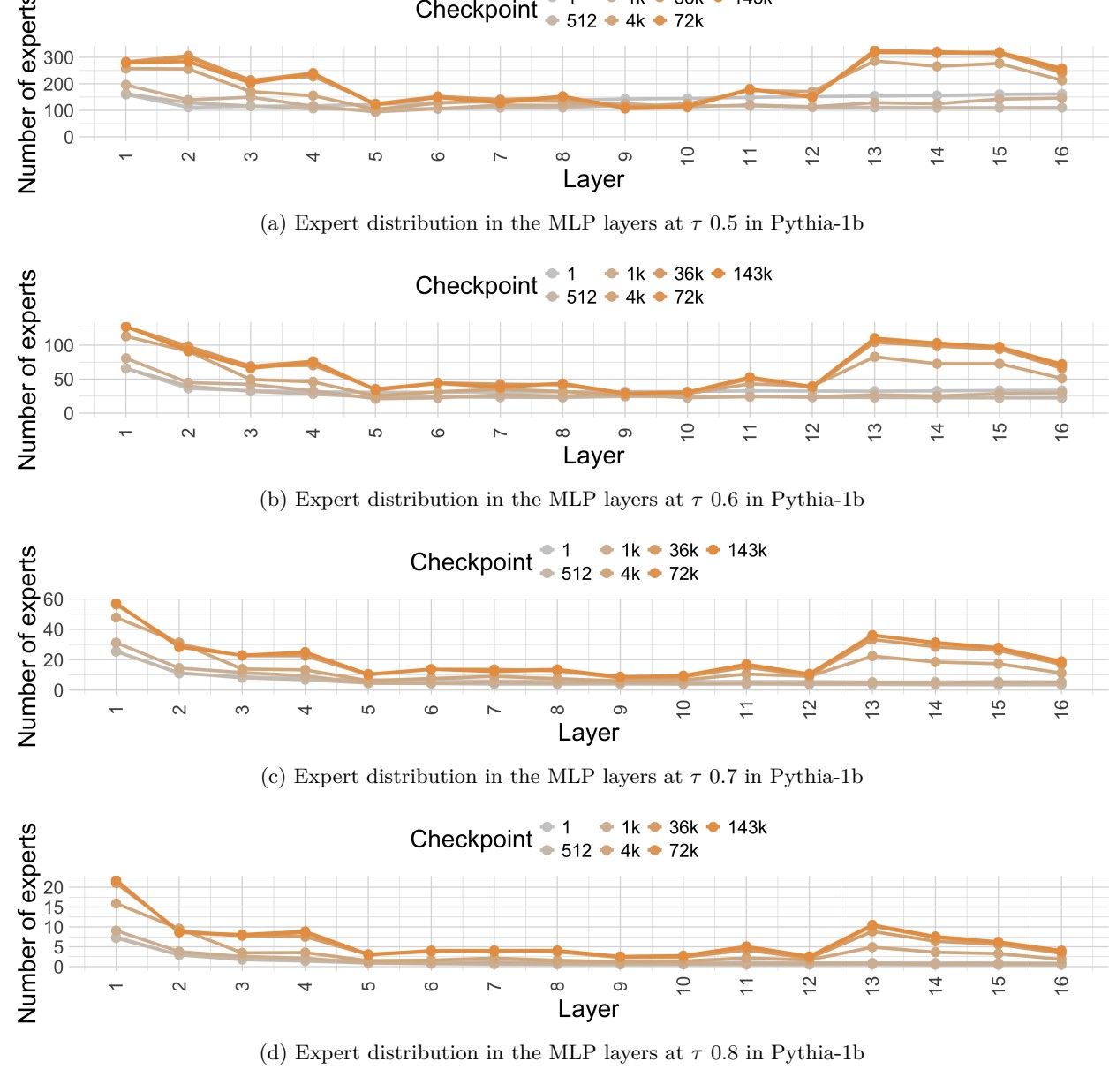

(a) Expert distribution in the MLP layers at $\tau$ 0.5 in Pythia-1b

(b) Expert distribution in the MLP layers at $\tau$ 0.6 in Pythia-1b

(c) Expert distribution in the MLP layers at $\tau$ 0.7 in Pythia-1b

(d) Expert distribution in the MLP layers at $\tau$ 0.8 in Pythia-1b

(e) Expert distribution in the MLP layers at $\tau$ 0.9 in Pythia-1b

Figure 19: The distribution of experts across in MLP layers in the Pythia-1b as a function $\tau$.

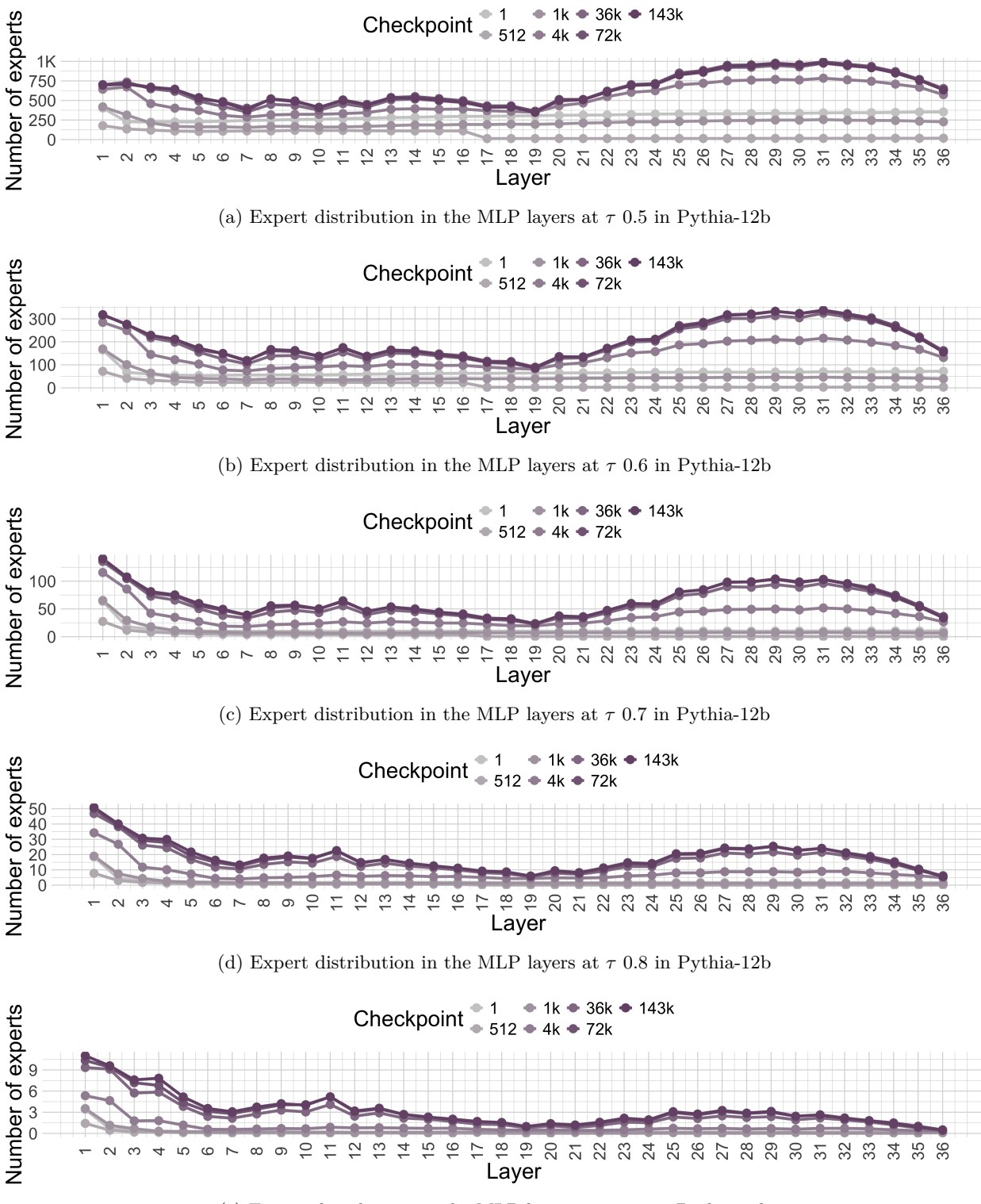

(a) Expert distribution in the MLP layers at $\tau$ 0.5 in Pythia-12b

(b) Expert distribution in the MLP layers at $\tau$ 0.6 in Pythia-12b

(c) Expert distribution in the MLP layers at $\tau$ 0.7 in Pythia-12b

(d) Expert distribution in the MLP layers at $\tau$ 0.8 in Pythia-12b

(e) Expert distribution in the MLP layers at $\tau$ 0.9 in Pythia-12b

Figure 20: The distribution of experts across in MLP layers in the Pythia-12b as a function $\tau$.

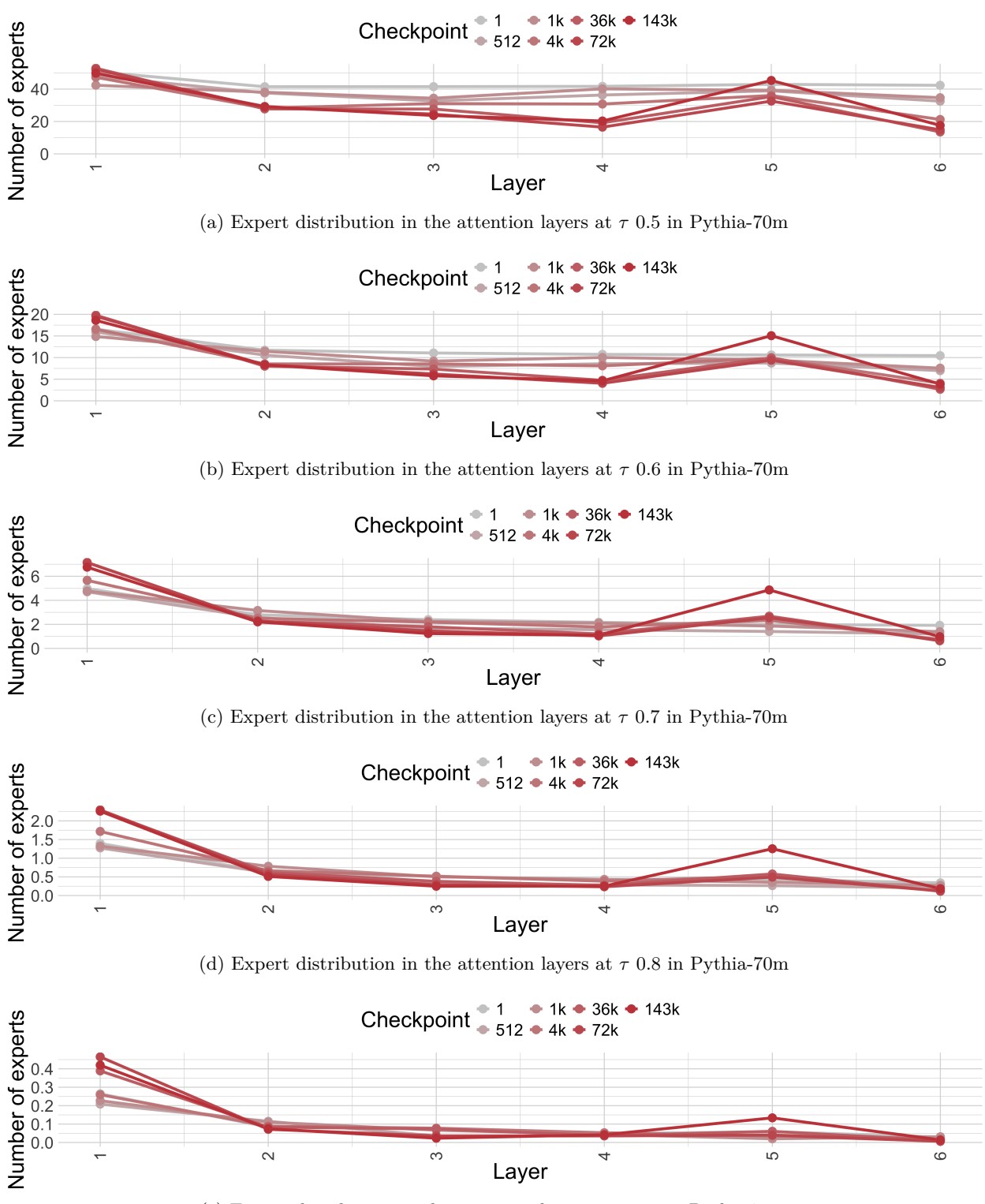

(a) Expert distribution in the attention layers at $\tau$ 0.5 in Pythia-70m

(b) Expert distribution in the attention layers at $\tau$ 0.6 in Pythia-70m

(c) Expert distribution in the attention layers at $\tau$ 0.7 in Pythia-70m

(d) Expert distribution in the attention layers at $\tau$ 0.8 in Pythia-70m

(e) Expert distribution in the attention layers at $\tau$ 0.9 in Pythia-70m

Figure 21: The distribution of experts across in attention layers in the Pythia-70m as a function $\tau$.

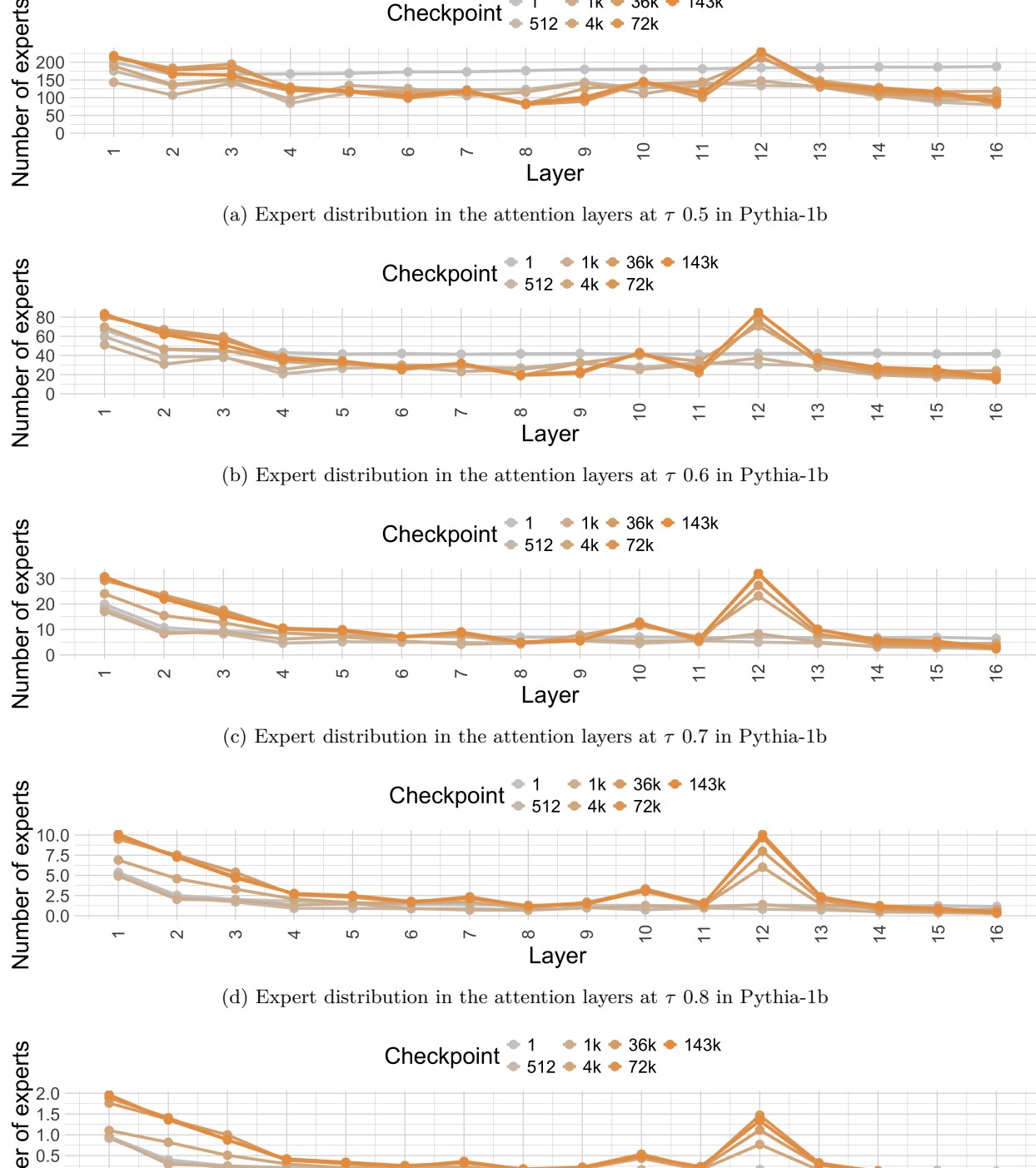

(a) Expert distribution in the attention layers at $\tau$ 0.5 in Pythia-1b

(b) Expert distribution in the attention layers at $\tau$ 0.6 in Pythia-1b

(c) Expert distribution in the attention layers at $\tau$ 0.7 in Pythia-1b

(d) Expert distribution in the attention layers at $\tau$ 0.8 in Pythia-1b

(e) Expert distribution in the attention layers at $\tau$ 0.9 in Pythia-1b

Figure 22: The distribution of experts across in attention layers in the Pythia-1b as a function $\tau$.

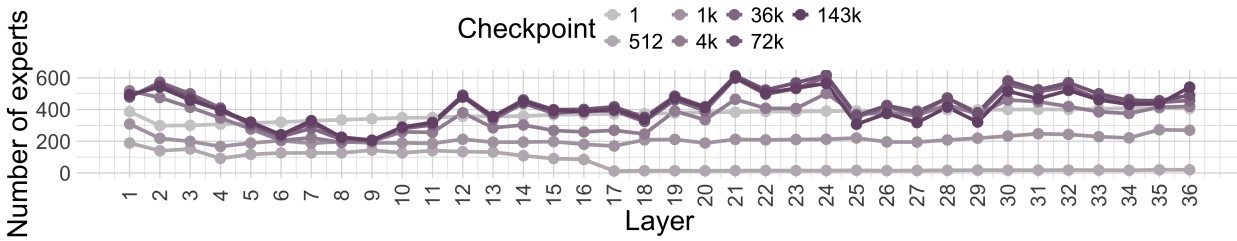

(a) Expert distribution in the attention layers at $\tau$ 0.5 in Pythia-12b

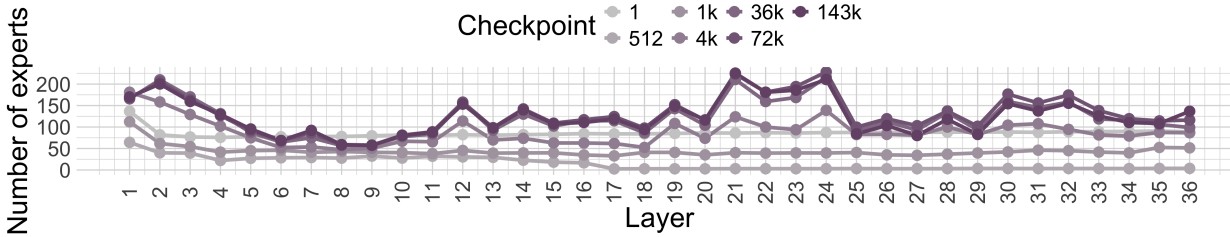

(b) Expert distribution in the attention layers at $\tau$ 0.6 in Pythia-12b

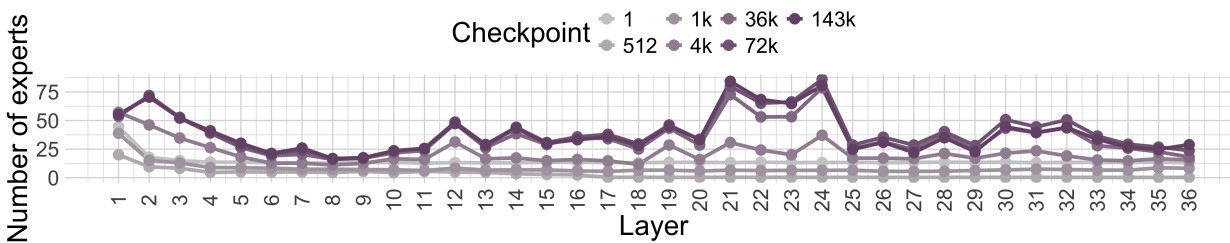

(c) Expert distribution in the attention layers at $\tau$ 0.7 in Pythia-12b

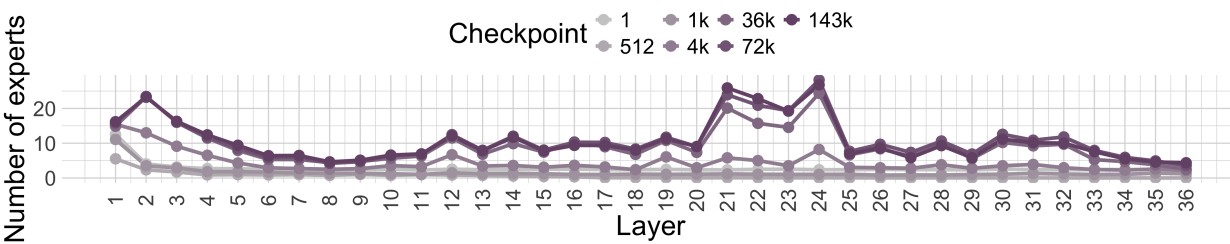

(d) Expert distribution in the attention layers at $\tau$ 0.8 in Pythia-12b

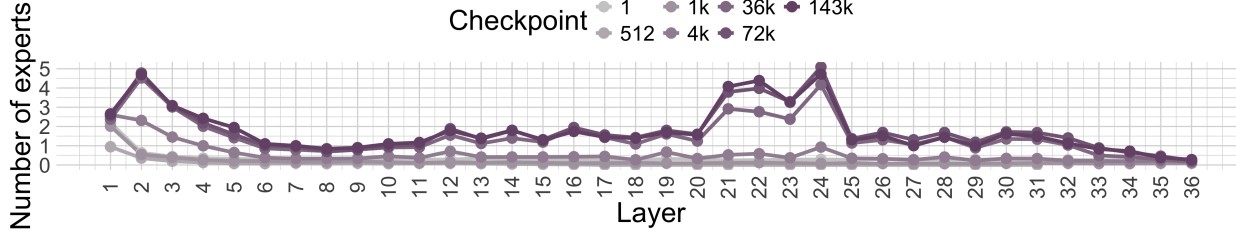

(e) Expert distribution in the attention layers at $\tau$ 0.9 in Pythia-12b

Figure 23: The distribution of experts across in attention layers in the Pythia-12b as a function $\tau$.

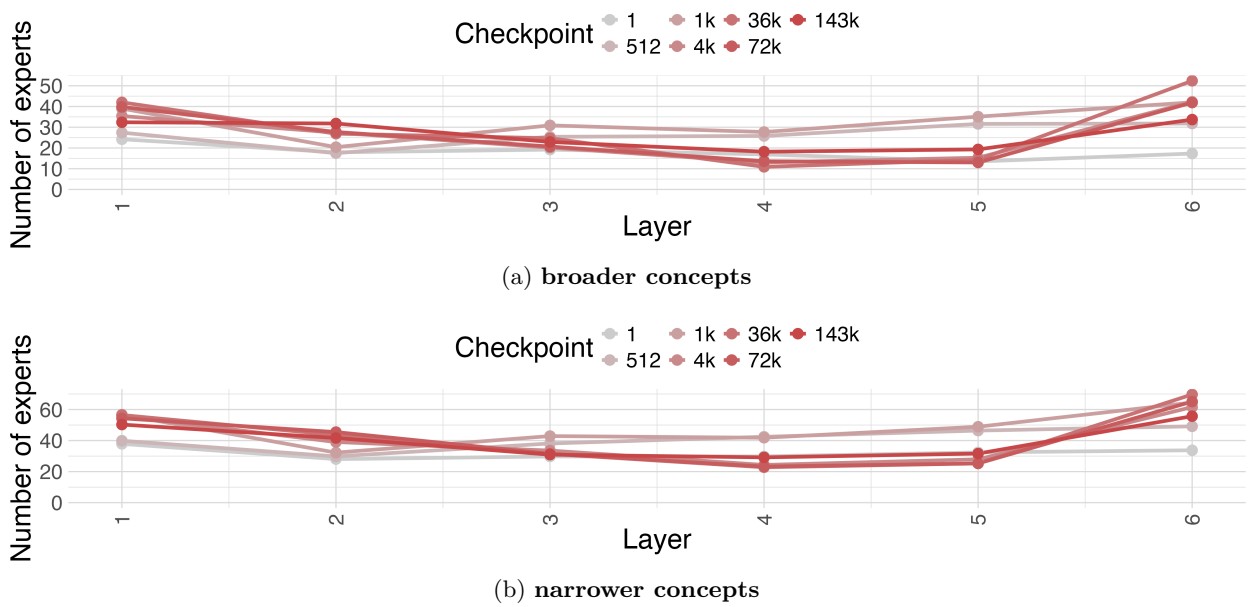

(a) **broader concepts**

(b) **narrower concepts**

Figure 24: Average number of experts identified for **broader concepts** (top) and **broader concepts** (bottom) in MLP layers at different depths, for different checkpoints of Pythia-70m.

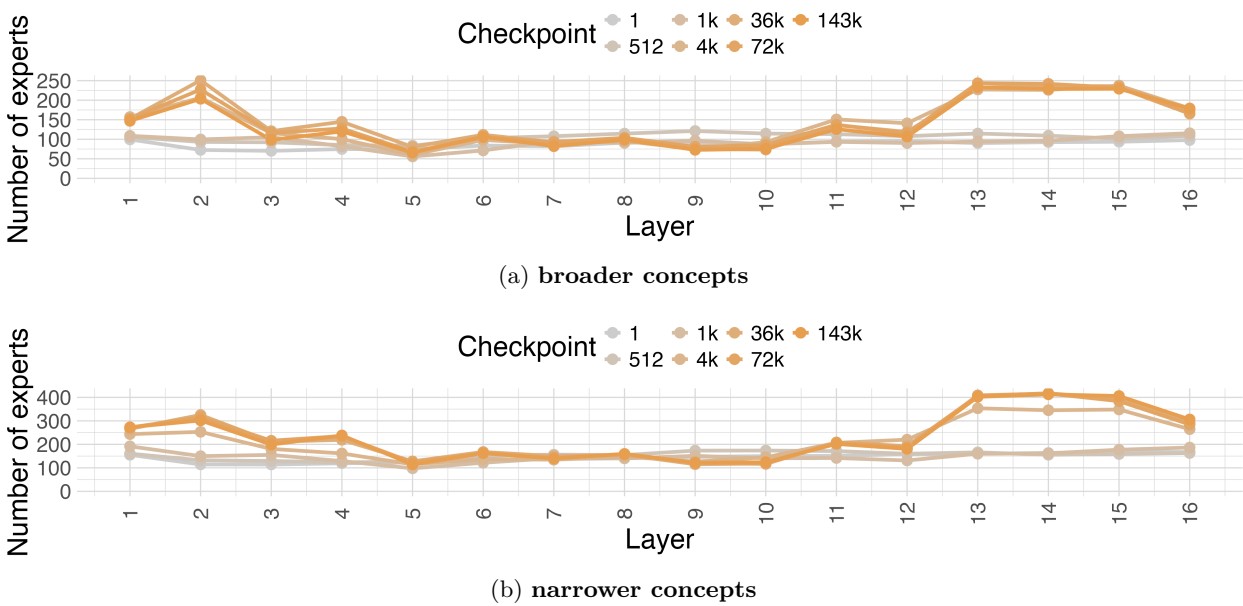

(a) **broader concepts**

(b) **narrower concepts**

Figure 25: Average number of experts identified for **broader concepts** (top) and **broader concepts** (bottom) in MLP layers at different depths, for different checkpoints of Pythia-1b.

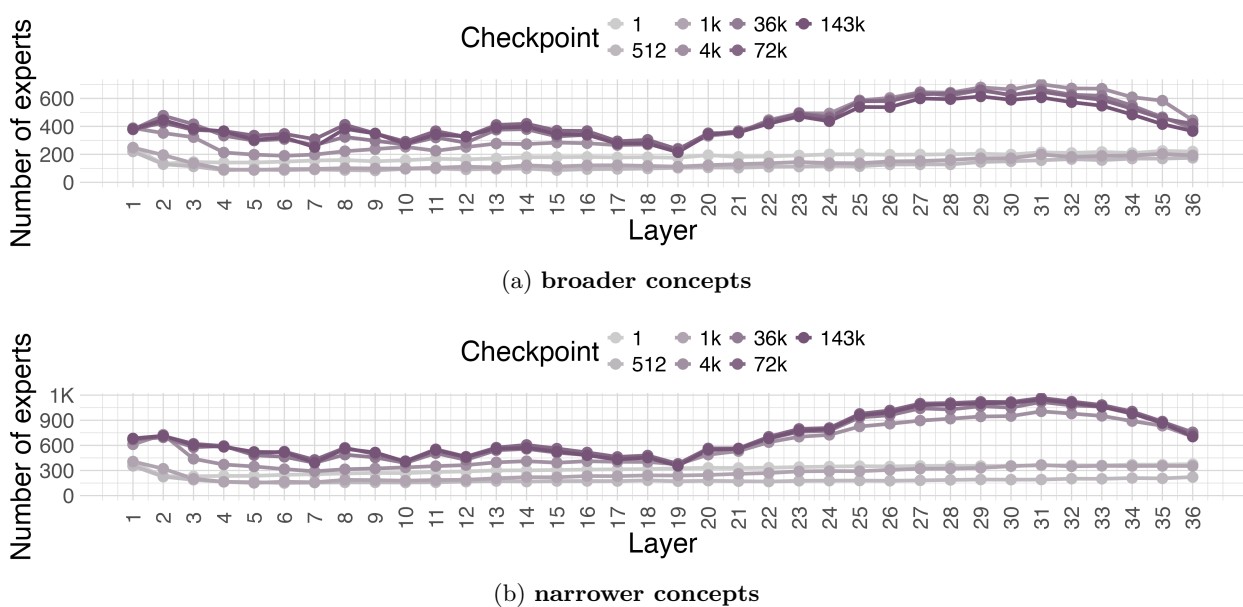

Figure 26: Average number of experts identified for **broader concepts** (top) and **broader concepts** (bottom) in MLP layers at different depths, for different checkpoints of Pythia-12b.

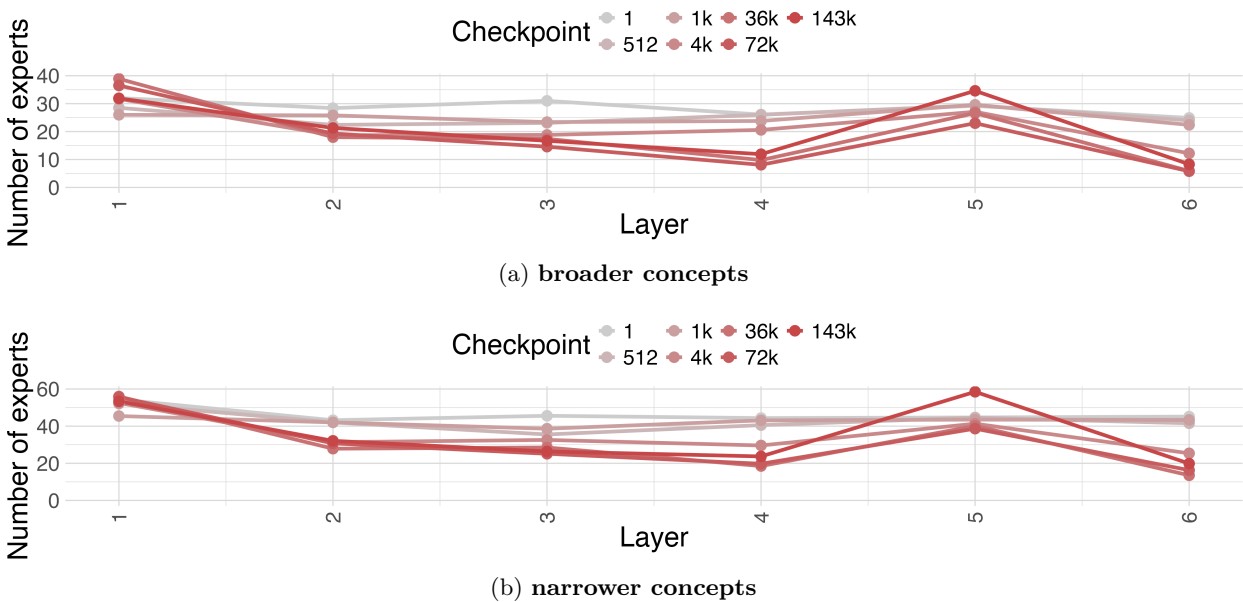

Figure 27: Average number of experts identified for **broader concepts** (top) and **broader concepts** (bottom) in the attention layers at different depths, for different checkpoints of Pythia-70m.

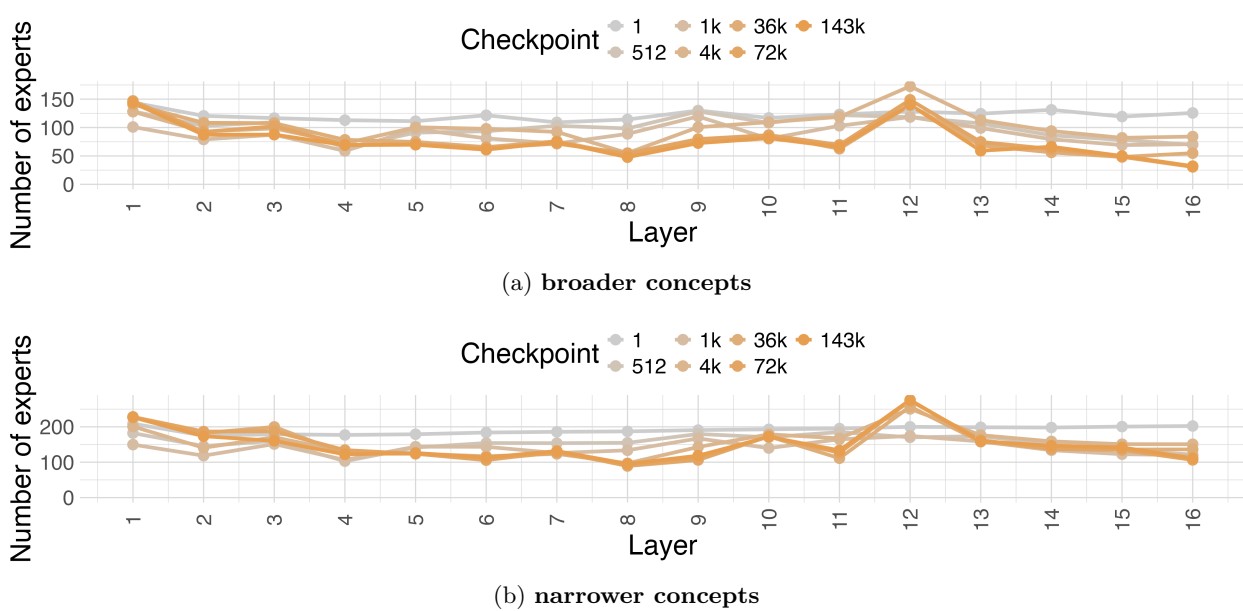

Figure 28: Average number of experts identified for **broader concepts** (top) and **broader concepts** (bottom) in the attention layers at different depths, for different checkpoints of Pythia-1b.

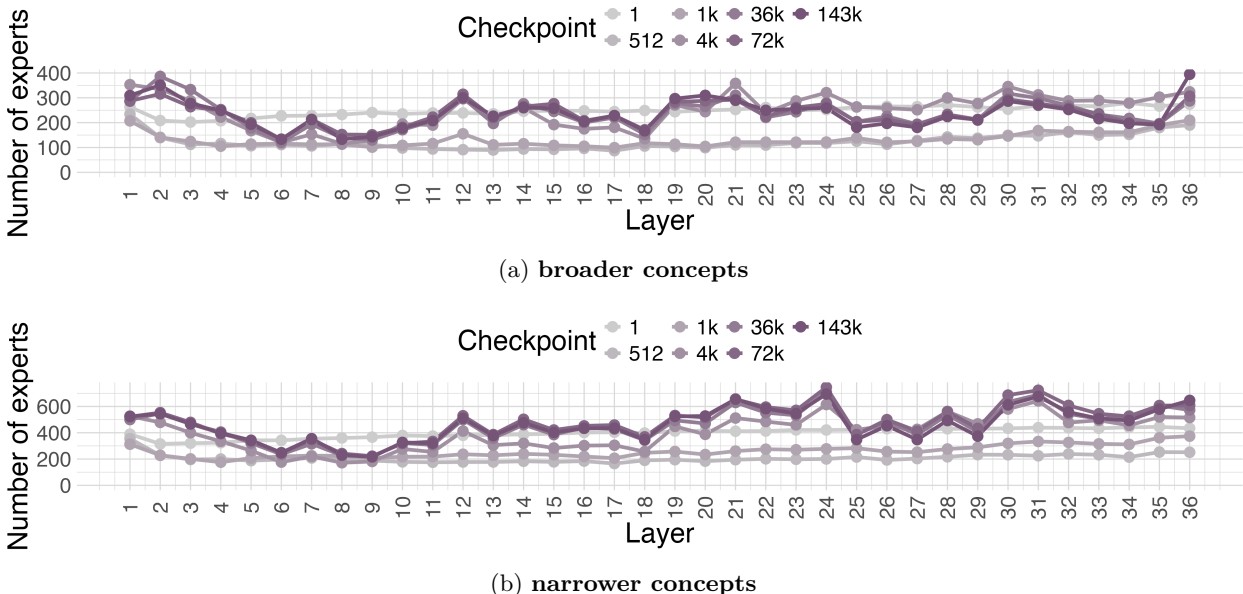

Figure 29: Average number of experts identified for **broader concepts** (top) and **broader concepts** (bottom) in the attention layers at different depths, for different checkpoints of Pythia-12b.

## H    Distribution of AP values for the expert neurons shared and not shared between the concepts in a pair

In Fig. 9 in Sec. 4.4, we showed that there is no difference in the raw AP values depending on whether the experts are shared by two concepts in a pair or not in Pythia-12b. Below, we provide evidence that this observation hold for smaller models too (see Fig. 30 for Pythia-70m and Fig. 31 for Pythia-1b).

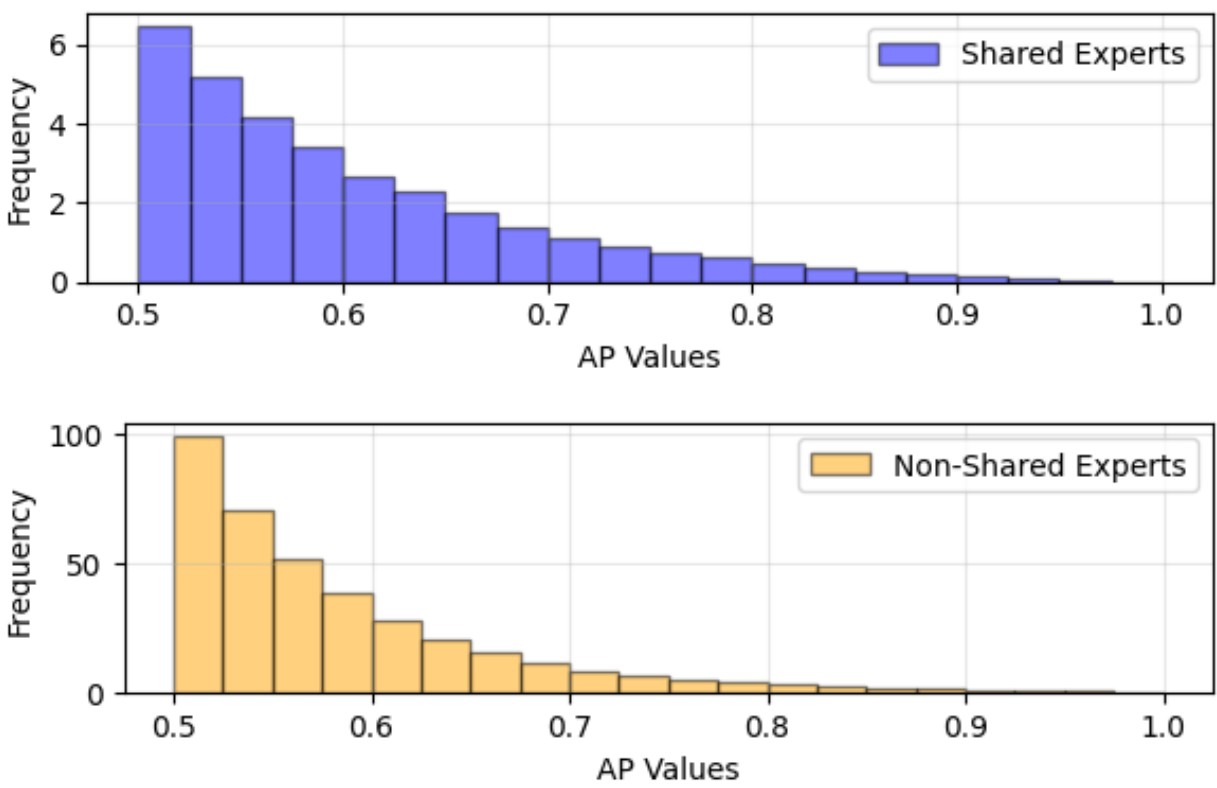

Figure 30: **Pythia 70m.** Histograms of raw AP values for the experts shared (blue) and not shared (yellow) between the concepts in a pair at checkpoint $143,000$.

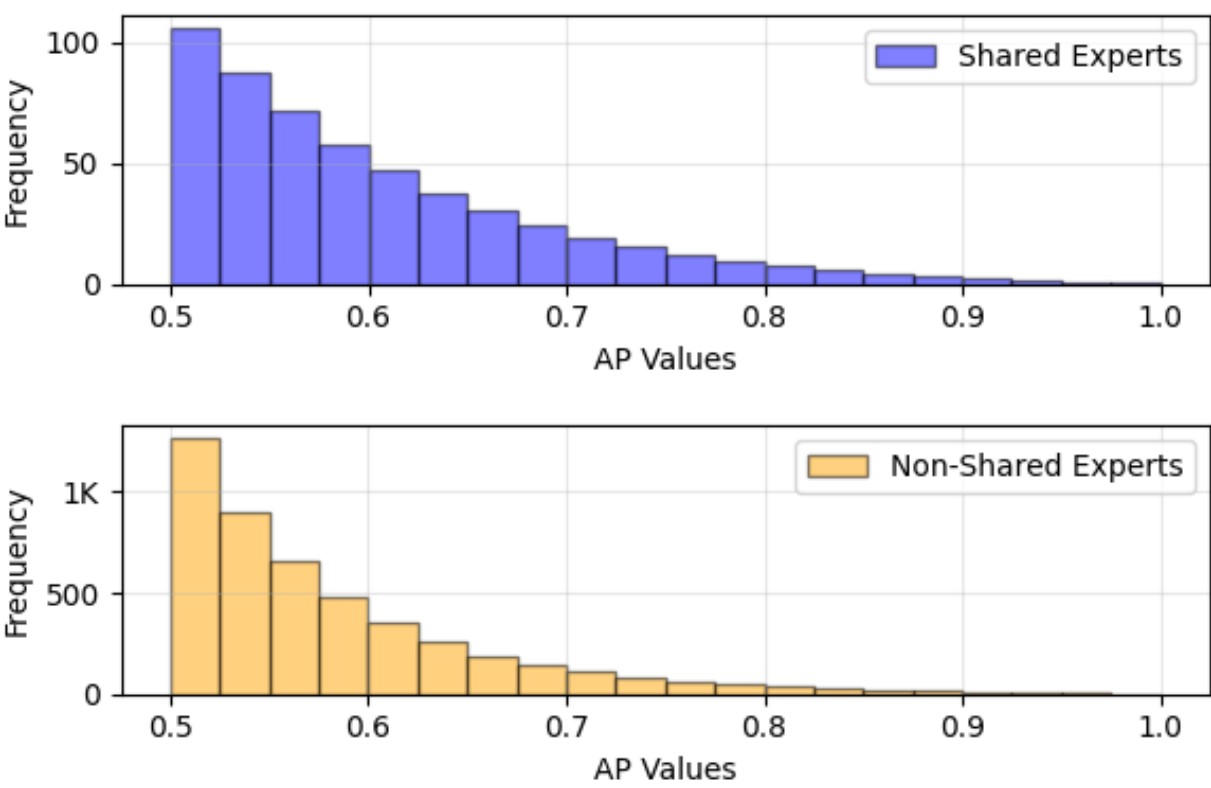

Figure 31: **Pythia 1b.** Histograms of raw AP values for the experts shared (blue) and not shared (yellow) between the concepts in a pair at checkpoint $143,000$.

## I   Computational budget

The concept dataset was parallelized over 8 A100 GPUs (80GB). Expert extraction took about 136 seconds per concept for the 12b Pythia model; about 27 seconds per concept for the 1b Pythia model; about 8 seconds per concept for the 70m Pythia model; and about 25 seconds per concept for GPT-2.

## J   License and Attribution

The MEN dataset used in this work is released under Creative Commons Attribute license. The SPP dataset is publicly available and used with permission from the authors. The pre-trained models are supported by public licenses the Pythia Scaling Suite (Apache), Mistral (Apache), GPT-2 (MIT), Gemma (gemma). GPT-4 is supported a proprietary license. We use an internal 80b-chat model and are unable to provide license information on it at this time.

