# OpenReview forum: "ExpertLens: Activation steering features are highly interpretable"
_TMLR — Accepted by TMLR_

### Review · Reviewer_6Grd · 2026-03-04

**Summary Of Contributions:**

## Summary

In this paper, the authors investigate the interpretability of expert neurons in neural networks. Those expert neurons are obtained via an exising activation steering method called *Finding Experts*. Those neurons when activated, steer the model's behavior in a specific direction, and the authors analyze their activation patterns and how they relate to perceived concept similarity in humans. The author propose ExpertLens, a method to test hypotheses about model representation. For a given concept (e.g., "cat"), the ExpertLens representation is the set of neurons whose activations are highly predictive of whether that concept is present in the input, together with (optionally) their predictive scores.

As a result of their analysis, the authors show that ExpertLens representations 1) capture concept representation in LLM, 2) are stable across datasets and models, 3) align closely with human representations. The authors also show how ExpertLens representations evolves with model training and model capacity.


## Strengths

This work addresses an important topic in Machine Learning: how to interpret the internal representations of neural networks. The proposed method, ExpertLens, is a novel approach to analyze the activation patterns of expert neurons and their relationship to a given perceived concept. The paper provides a comprehensive analysis of the interpretability of expert neurons and their alignment with human representations. The experiments are well-designed and provide strong evidence for the claims made in the paper. The paper is well-written and clearly presents the motivation, methodology, and results of the study.

One strength not explicitly mentioned in the paper is that the ExpertLens representations is robust to different activation functions (ReLU, GeLU, SwiGLU, etc.), layerwise scaling differences, training-stage differences, and model-size differences.
At first glance, when reading about "neurons as classifiers", one might think that the method implicitly assumes bounded activations. However, the ExpertLens representations are actually based on the relative ordering, not absolute scale. This means it works neuron-by-neuron, never compares raw activations across neurons, and never interprets magnitude as meaning.


## Weaknesses

The paper could be strengthened by providing more details about how to use to ExpertLens representations in downstream tasks, and how to use them to steer the model's behavior in a specific direction. The authors briefly touch on the steering model generation topic in section 4.1 but it did not involve the ExpertLens representations, only the top-500 expert neurons. For instance, what is would be the effect of steering only the expert neurons with $tau$ above a certain threshold?

The current approach relies on the assumption that concept can be captured by *individual* neurons as opposed to joint activations. The authors do acknowledge this limitation, however. I believe this is a common assumption in the interpretability literature, but it would be interesting to see how the ExpertLens representations could be extended to capture joint activations of neurons in the future.

**Additional Comments:**

I have used LLM to help shape the sentences for this review but all ideas, suggestions, and comments are my own.

**Audience:**

Yes

**Audience Explanation:**

Definitely, this paper addresses a fundamental question in the interpretability of neural networks, which is of great interest to researchers and practitioners in the field. The findings of this paper provide insights into how expert neurons capture concept representations, where are those neurons located and when they emerge during training. These insights can inform the development of more interpretable models and techniques for analyzing neural networks, which is a topic of broad interest in the machine learning community.

**Broader Impact Concerns:**

The paper does not explicitly discuss broader impact concerns, but the findings of this paper could have implications for the development of more interpretable models and techniques for analyzing neural networks. This could lead to increased transparency and accountability in AI systems, which is a positive outcome. However, it is also important to consider potential negative impacts, such as the misuse of interpretability techniques to manipulate or deceive users, and steering models in harmful directions. It is crucial for researchers and practitioners to be mindful of these potential risks and to use interpretability techniques responsibly.

**Claims And Evidence:**

Yes

**Claims Explanation:**

Yes, all claims made in the submission are supported by a dedicated experiments on multiple datasets, models and thresholds. The authors also reports standard deviation and confidence intervals for their results, which adds to the credibility of their findings. The paper is well-structured and clearly presents the evidence for each claim, making it easy for readers to follow the logic and understand the results.

**Requested Changes:**

The following are minor comments and typos. For my part, I don't have any critical changes that would prevent me to recommend this paper for acceptance.

### Minor

The term "expert" might be misleading. Reading the title at first, I thought "Expert" referred to MoE (Mixture of Experts) type of experts, but it actually refers to "expert neurons" obtained via an existing activation steering method called *Finding Experts*. I'm not asking to change anything, I just wanted to point that out in case you were hesitating between alternative names. I do think it becomes clear after reading the abstract and introduction, though.

In section 3.1, the sentence "To activate the expert neurons, their activations are set to their mean value over the positive set." seems out of place. Is setting the activation only use in the section 4.1 to validate the steering effects of those neurons during generation?

In section 3.3, what does "We use GPT-2 to select hyper-parameters." means concretely? Do the authors query GPT-2 to generate the value for those hyper-parameters?

At the end of section 3.2 (on page 4), the use of abbreviation is not consistent, e.g. (Section 4.3 and Sec. 4.1)

On page 4, last paragraph there's a em dash followed by an hyphen. Also, there's a long whitespace in "App.   B"

On page 5, the formatting of the title could be improved (i.e., adding space before "Overlap")

On page 6, in section 4.3, there's a very long sentence that could be split for better readability, "We considered several more...".

On page 12, typo "we have the have the best..."

In App. A, is it expected that for GPT-4 generated stories, changing the prompt (Fact vs. Story) doesn't change the generation?

On page 18, in App. B. What is the different between the bottom two generations? Both their titles are the same.

Still in App. B, I'm curious to know if the authors have tried to force deactivating the expert neurons for a given concept while asking to generate a text about that concept?

Page 25, in Table 2, the last value of the table has a different font size than the rest of the table.

---

> ### Author Response · Authors · 2026-03-20
> **Reply to Reviewer 6Grd**
>
> >The paper could be strengthened by providing more details about how to use to ExpertLens representations in downstream tasks, and how to use them to steer the model's behavior in a specific direction. The authors briefly touch on the steering model generation topic in section 4.1 but it did not involve the ExpertLens representations, only the top-500 expert neurons. For instance, what is would be the effect of steering only the expert neurons with above a certain threshold?
>
> We haven’t done a systematic exploration of this but we have attempted to activate larger sets of neurons in early exploration. Steering is a balancing act — one the one hand, we’d would want to induce a concept, on the other hand we don’t want to break other model abilities in the process. The more neurons one activates, the more it degrades other model’s abilities. We have attempted to activate a significantly larger number of neurons (which would correspond to a lower threshold). Once the set of activated neurons gets large, it significantly degrades model perplexity and leads to low-quality generations. To us, this illustrates the dichotomy between steering and interpretability — what is good for one aspect, isn’t necessarily good for the other.
>
> >Requested Changes:
>
> >The term "expert" might be misleading. Reading the title at first, I thought "Expert" referred to MoE (Mixture of Experts) type of experts, but it actually refers to "expert neurons" obtained via an existing activation steering method called Finding Experts. I'm not asking to change anything, I just wanted to point that out in case you were hesitating between alternative names. I do think it becomes clear after reading the abstract and introduction, though.
>
> Thank you for bringing this to our attention. We’re somewhat attached to ExpertLens if we’re entirely honest and we’ll keep the name.
>
> >In section 3.1, the sentence "To activate the expert neurons, their activations are set to their mean value over the positive set." seems out of place. Is setting the activation only use in the section 4.1 to validate the steering effects of those neurons during generation?
>
> Yes. We have now rephrased this in Section 3.1: “Additionally, we establish the causal connection between the expert neurons and the expression of the concept in the generations by activating the top-500 experts (i.e., by setting their activations to the mean value over the positive set).”
>
> >In section 3.3, what does "We use GPT-2 to select hyper-parameters." means concretely? Do the authors query GPT-2 to generate the value for those hyper-parameters?
>
> We have now clarified this: “We use GPT-2 to select hyper-parameters discussed in Fig. 1 (e.g., the size of a positive and negative datasets) and validate that our data identifies a stable set of experts”
>
> >At the end of section 3.2 (on page 4), the use of abbreviation is not consistent, e.g. (Section 4.3 and Sec. 4.1)
>
> Thank you for catching this. We went through the paper and addressed these inconsistencies throughout the paper.
>
> >On page 6, in section 4.3, there's a very long sentence that could be split for better readability, "We considered several more...".
>
> We now split the sentence into 2 for readability.
>
> >In App. A, is it expected that for GPT-4 generated stories, changing the prompt (Fact vs. Story) doesn't change the generation?
>
> Great catch! It’s a copy-and-paste error on our part. We’ve updated the manuscript with correct examples — the generations do change between the fact and story prompts for all models.
>
> >On page 18, in App. B. What is the different between the bottom two generations? Both their titles are the same.
>
> They are indeed the same. We’ve clarified in the paper: ‘For the intervened model, we provide two sample generations to illustrate the variability.’
>
> >Still in App. B, I'm curious to know if the authors have tried to force deactivating the expert neurons for a given concept while asking to generate a text about that concept?
>
> Great question! We have not. But Suau et al. (2024) explores this somewhat. Specifically, they found that while it’s relatively straightforward to ‘activate’ a concept by setting its experts to their expected value, it’s quite a bit harder to ‘deactivate’ a concept by zero-ing out its experts and it requires more complex approaches such as applying some deactivation to the entire network. We hypothesize that it has to do precisely with the concept overlap that we observe in ExpertLens — expert neurons are shared across related concepts. Therefore deactivating the neurons for one concept may not lead to its deletion since the experts for related concepts also contain some information about the target concept.
>
> Suau, X., Delobelle, P., Metcalf, K., Joulin, A., Apostoloff, N., Zappella, L., Rodriguez, P. Whispering Experts: Neural Interventions for Toxicity Mitigation in Language Models, ICML 2024
>
> >Typos/inconsistencies
>
> We have fixed all typos and inconsistencies pointed out by the reviewer.

---

> > ### Comment · Reviewer_6Grd · 2026-03-24
> >
> > Thank you for the clarifications and the additional comparison with SAEs.
> >
> > >> The paper could be strengthened by providing more details about how to use to ExpertLens representations in downstream tasks, and how to use them to steer the model's behavior in a specific direction. The authors briefly touch on the steering model generation topic in section 4.1 but it did not involve the ExpertLens representations, only the top-500 expert neurons. For instance, what is would be the effect of steering only the expert neurons with above a certain threshold?
> >
> > > We haven’t done a systematic exploration of this but we have attempted to activate larger sets of neurons in early exploration. Steering is a balancing act — one the one hand, we’d would want to induce a concept, on the other hand we don’t want to break other model abilities in the process. The more neurons one activates, the more it degrades other model’s abilities. We have attempted to activate a significantly larger number of neurons (which would correspond to a lower threshold). Once the set of activated neurons gets large, it significantly degrades model perplexity and leads to low-quality generations. To us, this illustrates the dichotomy between steering and interpretability — what is good for one aspect, isn’t necessarily good for the other.
> >
> > Any ideas on how to use to *ExpertLens representations* for downstream tasks? Maybe I misunderstood it, but the "ExpertLens representations" is different from just the "top-500 expert neurons", right?

---

> > > ### Author Response · Authors · 2026-03-24
> > >
> > > >Any ideas on how to use to ExpertLens representations for downstream tasks? Maybe I misunderstood it, but the "ExpertLens representations" is different from just the "top-500 expert neurons", right?
> > >
> > > Yes, your understanding is correct — top-500 experts include the highest quality experts only (on average the neurons above the expertise threshold of 0.9 and a subset of neurons above the expertise threshold of 0.8 of the ExpertLens representations).
> > >
> > > We see the primary use of ExpertLens representations for model interpretability. We discuss some of the potential use cases in the Discussion (p. 12):
> > >
> > > "We see potential uses for this approach as a tool for studying representational alignment in a variety of domains. Given our definition of a concept as a set of examples, it can be readily extended to more abstract concepts like safety, toxicity or value alignment. For instance, in safety alignment, one could ask questions such as: do existing alignment methods truly make the representations more aligned? There is an ‘alignment tax’ associated with alignment meaning that, after applying safety alignment, model performance drops on other tasks (Askell et al., 2021). Is this because other aspects of the representation become misaligned? Understanding these questions could lead to improved alignment, while providing insight into how to mitigate the undesirable consequences of applying changes to model representation. Going beyond alignment, ExpertLens could be a promising tool to study the relationship between the training data and knowledge representation in the model, which could guide us to design better synthetic datasets."

---

> > > > ### Comment · Reviewer_6Grd · 2026-03-24
> > > >
> > > > My bad, I missed that. Thanks for the pointer.

---

### Review · Reviewer_RiZP · 2026-03-08

**Summary Of Contributions:**

Summary
The paper introduces "ExpertLens", a method designed to find interpretable representations from the neurons identified by activation steering. The steering method used is the "finding experts" method of Suau et al., 2023. "ExpertLens" treats the set of expert neurons associated with a concept as its representation. They measure inter-concept similarity via Jaccard overlap of expert sets.

The paper has three main claims: (1) expert neuron sets are stable across dataset variations and model choices; (2) the Jaccard similarity between expert sets correlates with human-perceived concept similarity (on the MEN and SPP datasets), outperforming embedding-based baselines; and (3) expert representations evolve meaningfully over training and scale with model capacity.

Strengths:

The core question, namely whether activation steering features are interpretable, is an important question, having a bearing both on mechanistic interpretability and steering/control methods.

The stability analysis in section 4.2 is comprehensive, sweeping over multiple dataset sizes, generation models, and fold splits, and provides useful practical guidance (e.g., that 400 positive sentences is good enough).

The comparison to human similarity judgments using established psycholinguistic benchmarks (MEN, SPP) is a nice evaluation strategy, and the correlation with inter-human agreement (~0.84) provides context on what is possible.

The concept organization analysis (Figure 5) is quite striking, providing a strong demonstration that expert overlap captures domain-level structure.

The training dynamics analysis (Section 4.4) offers interesting observations about how experts emerge, stabilize, and specialize over the course of training, although the implications here are a bit more speculative.

Weaknesses:

The novelty is limited: the finding experts method is entirely from the prior work, and the primary contribution is applying Jaccard similarity to expert sets and correlating with human judgments, which is conceptually straightforward.

The evaluation is restricted to simple, concrete, single-word concepts (animals, colors, furniture), making it unclear whether the approach generalizes to more interesting cases. This is especially limiting given that the prior work already demonstrated that expert neurons can handle quite nuanced concepts and distinctions (e.g., "lead" the metal vs. being ahead) and that experts can be used with the abstract concept of toxicity. It would be very interesting to study a few more complicated concepts and see if the alignment between human and LLM concepts still holds.

The embedding baselines are relatively weak: only single-word embeddings and averaged sentence embeddings are compared. More recent baselines such as SAEs are absent.

The experiments are conducted exclusively on Pythia models (70m–12b), which are quite small and dated as of 2026. The Gemma results are relegated to an appendix and not discussed in depth.

**Audience:**

Yes

**Audience Explanation:**

The paper addresses a question that sits between two active research areas: mechanistic interpretability and activation steering.

The finding that activation steering features align with human conceptual structure is relevant to interpretability researchers. The practical implication could be useful to practitioners who want to understand what their models encode without the computational cost of training SAEs or probes.
That said, the scope of the findings (limited to concrete, single-word concepts in relatively old, small models) somewhat limits the amount of interest. If extended to more complex representations, such as the toxicity representation already studied in the prior work, there would be a lot more interest.

**Broader Impact Concerns:**

No ethical implications

**Claims And Evidence:**

Yes

**Claims Explanation:**

As noted above, the paper makes three claims: stability of expert neurons, Jaccard similarity correlates with human concept similarity, and expert representations evolve over the course of training.

Stability: This is supported well. The pilot study in Section 4.2 is well-designed, with thorough variation of dataset size, generation model, and random splits. The high within-concept Jaccard overlap ($\tilde 0.8$ at $tau=0.5$) versus near-zero inter-concept overlap is convincing evidence of stability. However, the stability decreases substantially at higher $\tau$ values, and the authors don't really engage with this very much. I wonder if the authors could use a statistic that doesn't depend on the expert size in a confusing way.

Alignment with humans: This is supported reasonably well. The correlations of 0.70–0.79 with similarity judgments are high, especially when inter-human agreement is 0.84.

The MEN dataset contains mostly concrete nouns, which are arguably the easiest case for any representation method. The generalization to more abstract or complex concepts is untested. The main methodological issue is that the embeddings are relatively outdated. Methods such as SAEs would provide a comparison that is more relevant for practitioners choosing between options in 2026. The SPP analysis is a bit less convincing, with only three coarse similarity bins, a much less sensitive test than the continuous MEN correlations. This is a limitation that the authors address clearly in the text. It would also be nice to see more complex concepts than the concrete nouns used here, although this doesn't affect how supported the more limited claims made by the authors are.

Training dynamics: This is mostly well supported. The observations about expert emergence over training are quite interesting data points, but the authors' inferences from the observations are not very clearly supported. For instance, the authors say that the observations are expected due to the information bottleneck framework, but don't make this concrete or study it further.

The claim that "more specialized experts take longer to learn" is demonstrated by Figure 7 showing that higher-$\tau$ results in some models, such as 70m, having less overlap earlier on in training. But this relies on identifying 'specialization' with higher-$\tau$, which is not very clearly described in the text.

**Requested Changes:**

Non-Critical (would strengthen the paper but not required)

Stronger baselines. The embedding baselines are quite weak. The paper would be improved if the authors compared to SAEs, such as those already available for the Gemma models via GemmaScope.

Controlling for expert set size in Jaccard similarity. As the authors note, the Jaccard similarity is sensitive to set size: two concepts with very large expert sets will tend to have higher overlap by chance than two concepts with small expert sets. Is there a statistic which doesn't have this flaw, such as a normalized variant of the similarity?

Broader range of concepts. The restriction to single-word concrete nouns is a limitation, especially as the prior work has already demonstrated that experts can be found for both nuanced concepts ('lead' vs 'lead') and complex concepts such as toxicity. The authors could extend the work to more interesting concepts with more real-world applicability.

More modern model architectures. The reliance on Pythia models limits the insights' applicability. It would strengthen the work if the Gemma results in the appendix were brought into the main text and discussed more thoroughly.

The discussion in section 4.4 could be rephrased to overstate less. The 'this is expected' phrasing is not really supported by any evidence, and writing 'this could be explained via' would be better.

---

> ### Author Response · Authors · 2026-03-20
> **Reply to Reviewer RiZP**
>
> >Weaknesses
> >The novelty is limited: the finding experts method is entirely from the prior work, and the primary contribution is applying Jaccard similarity to expert sets and correlating with human judgments, which is conceptually straightforward.
>
> There are two non-trivial observations we would like to highlight. First, the concept similarity captured by ExpertLens is at human-level: for the LLMs we study, our method shows that model–human alignment effectively reaches the ceiling given by inter-annotator agreement. Second, we observe that model capacity plays only a small role in this alignment, which mirrors the findings in the vision domain (Muttenhaler et al. 2023) but hasn’t been demonstrated for language. Neither of these observations are trivial, nor have been reported in previous literature.
>
> Additionally, we now provide a comparison to SAEs (see the General Response), showing that our approach outperforms these features.
>
> >The experiments are conducted exclusively on Pythia models (70m–12b), which are quite small and dated as of 2026. The Gemma results are relegated to an appendix and not discussed in depth.
>
> We completely agree that Pythia is a bit dated. Our motivation for using Pythia was to have access to different model sizes trained on the same data with publicly available checkpoints to study expert neurons as function of model size and training progression. Olmo models (https://allenai.org/olmo) would have been a better choice. Unfortunately, we were not aware of them until the bulk of the data was already collected, making a switch prohibitively expensive in terms of compute.
>
> We now discuss the performance of ExpertLens on Gemma-2-2b in Section 4.3.
>
> >Requested Changes:
> >Stronger baselines. The embedding baselines are quite weak. The paper would be improved if the authors compared to SAEs, such as those already available for the Gemma models via GemmaScope.
>
> Thank you for suggesting this baseline. We now provide a comparison to SAEs (see the General Response), showing that our approach outperforms these features.
>
> >Controlling for expert set size in Jaccard similarity. As the authors note, the Jaccard similarity is sensitive to set size: two concepts with very large expert sets will tend to have higher overlap by chance than two concepts with small expert sets. Is there a statistic which doesn't have this flaw, such as a normalized variant of the similarity?
>
> We agree with the reviewer — Jaccard similarity is sensitive to the set size. We did not normalize it in this work because we do not think that it’s an issue in practice. Specifically, in Appendix D, we present the correlations between threshold-free metrics (cosine similarity over raw AP values and symmerized KL-divergence) with human judgments. While neither of these metrics depend on the set size as Jaccard similarity, both of these metrics show the same strength of the correlation with human judgements as Jaccard similarity, suggesting that while set size sensitivity is a consideration in principle, it is not a concern for interpreting our results.
>
> >Broader range of concepts. The restriction to single-word concrete nouns is a limitation, especially as the prior work has already demonstrated that experts can be found for both nuanced concepts ('lead' vs 'lead') and complex concepts such as toxicity. The authors could extend the work to more interesting concepts with more real-world applicability.
>
> This comment was addressed as part of our reply to Reviewer vULU.
>
> >More modern model architectures. The reliance on Pythia models limits the insights' applicability. It would strengthen the work if the Gemma results in the appendix were brought into the main text and discussed more thoroughly.
>
> Thank you for the suggestion. We now discuss the performance of ExpertLens on Gemma-2-2b in Section 4.3.
>
> >The discussion in section 4.4 could be rephrased to overstate less. The 'this is expected' phrasing is not really supported by any evidence, and writing 'this could be explained via' would be better.
>
> Point taken. We have adjusted the wording.

---

> > ### Author Response · Authors · 2026-04-06
> >
> > Dear Reviewer RiZP,
> >
> > As the discussion period is nearing the end, we would like to gently check in to see whether our responses have adequately addressed your concerns. If there are any remaining questions or concerns that would benefit from further discussion or clarification, please let us know. We would appreciate the opportunity to address them.
> >
> > Thank you!

---

### Review · Reviewer_vULU · 2026-03-09

**Summary Of Contributions:**

The paper studies whether features discovered by activation steering correspond to interpretable semantic concepts in large language models. Building on the “finding experts” method, the authors introduce ExpertLens, which represents a concept as the set of neurons most predictive of it. They analyze overlap between such expert neuron sets to measure semantic similarity between concepts. Experiments show that (1) expert neurons are relatively stable across datasets and models, (2) neuron-set similarity correlates with human semantic similarity judgments (e.g., MEN dataset), and (3) expert representations evolve during training and with model scale.

## Strengths:

- Clear and simple framework for analyzing activation-steering features.

- Empirical study across models and training checkpoints.

- Interesting observation that neuron-set similarity aligns with human semantic similarity.

## Weaknesses:

- Limited methodological novelty beyond prior “finding experts” activation-steering work.

- Experiments focus mainly on simple lexical concepts.

**Audience:**

Yes

**Audience Explanation:**

Interpretability of LLM representations and activation steering methods are active topics in the TMLR community. The paper’s perspective of representing concepts through sets of expert neurons is simple and may inspire further work on representation analysis.

However, the scope of experiments is relatively limited, and the contribution appears somewhat incremental relative to existing activation-steering and interpretability work.

**Broader Impact Concerns:**

None.

**Claims And Evidence:**

Yes

**Claims Explanation:**

The paper provides consistent empirical evidence for its claims. In particular, the authors show that expert neuron sets are stable across datasets and model sizes and that similarity between these sets correlates with human similarity judgments. The experimental setup and metrics are generally clear.

However, the interpretability claim is supported mainly through correlation with human similarity datasets, which provides indirect evidence. Additional causal tests or evaluations on more complex concepts would strengthen the conclusions.

**Requested Changes:**

See weaknesses.

---

> ### Author Response · Authors · 2026-03-20
> **Reply to Reviewer vULU**
>
> >Limited methodological novelty beyond prior “finding experts” activation-steering work.
>
> This depends on how one views a methodological contribution. Our intent with this work is not to provide a new activation steering method but rather to understand the potential of the current approaches for interpretability. The method for identifying neurons was introduced by Suau et al. (2023) in the context of activation steering research, i.e., with the goal of directing LLM outputs towards a desired direction (e.g., reducing toxicity). We investigate a novel application of this approach as a technique for model interpretability. Specifically, we show that the neurons discovered with this method provide reliable information about how models represent concepts. Hence, this is a reliable method to test model alignment. Our results show that the correlation between expert-based concept similarity measures and human concept similarity evaluations is comparable to inter-human correlation.
> Our contribution differs from that of Suau et al in its motivation and application. Specifically, our contributions are as follows:
>
> * we show that the method used to identify the experts is stable and reliable —previous work did not investigate whether an expert set was sufficient or reliable, but rather only explored whether the given set allowed to guide generation towards desired outputs
> * we show that the representations captured with this approach align closely with human representations, both at the level of concept similarity and in terms of concept organization
> * we provide an analysis of how human-model alignment evolves with model training and depends on model capacity
>
>  Additionally, we now provide a comparison to SAEs (see the General Response), showing that our approach outperforms these features.
>
> >Experiments focus mainly on simple lexical concepts.
>
> Analyzing multi-word and more complex concepts is indeed an interesting direction. In our work, we focused on single-word concepts because i) this is an important first step; ii) to the best of our knowledge, there is no publicly available dataset annotated with concept similarity for multi-word expressions. Would the reviewer be able to point us to one? Without such dataset, it is not clear how one could run an analysis of human-model alignment, and conducting a user study to collect human similarity judgments is beyond the scope of this paper.
> Of note: while there is no easy solution to analyze an alignment for multi-word concepts, we run our study on concepts drawn from two different datasets (MEN and SPP) to increase the coverage of concept types represented in the study.

---

> ### Author Response · Authors · 2026-04-06
>
> Dear Reviewer vULU,
>
> As the discussion period is nearing the end, we would like to gently check in to see whether our responses have adequately addressed your concerns. If there are any remaining questions or concerns that would benefit from further discussion or clarification, please let us know. We would appreciate the opportunity to address them.
>
> Thank you!

---

### Author Response · Authors · 2026-03-20
**General Response**

We thank the reviewers for their thoughtful comments and suggestions on improving the paper.

A common theme that emerged across multiple reviewers is the lack of more modern baselines such as sparse auto-encoders (SAEs) (Reviewer vULU, Reviewer RiZP) and a more explicit discussion of more modern model architectures, which in the current version of the paper takes place in the appendix. In response to these comments, we have expanded Section 4.3 of the paper to include a comparison between ExpertLens representations and SAE features for Gemma-2-2b extracted from Gemma Scope. In summary, we find that ExpertLens representations are more aligned with human similarity judgements that SAEs (highest Spearman correlations are 0.81 and 0.58 for ExpertLens and SAEs respectively). Our hypothesis is that due to the disentangled nature of SAE features, they are losing some of the concept associations captured by the ExpertLens, which matter for similarity. It is also possible that with a more thorough parameter sweep, SAEs would perform comparably to ExpertLens, which is essentially at the upper bound given that the human-human correlation is 0.84. ExpertLens will be preferred in that scenario as well since, unlike the SAEs, it doesn’t require training.

We provide the answers to the specific reviewer questions in the reply to the individual reviewers.

---

### Decision · Action_Editor_hmMN · 2026-04-01

**Recommendation:** Accept as is

**Additional Comments:**

All reviewers ultimately recommended acceptance.

**Audience:**

Yes

**Audience Explanation:**

This paper focuses on interpretability of LLMs, which is of clear interest to members of the TMLR community.

**Claims And Evidence:**

Yes

**Claims Explanation:**

Yes, all reviewers ultimately agreed that the evaluation was convincing. Experiments demonstrated that the "expert" neuron sets are stable across datasets and model sizes and that similarity between these sets correlates with human similarity judgments. Some initial concerns regarding baselines and the human evaluation setup were cleared up after rebuttal.